# A multi-omics integrative analysis based on CRISPR screens re-defines the pluripotency regulatory network in ESCs

Yan Ruan[1,9], Jiaqi Wang[1,2,9], Meng Yu[1,3,9], Fengsheng Wang[1,4], Jiangjun Wang[1,5], Yixiao Xu[1], Lianlian Liu[1], Yuda Cheng[1], Ran Yang[1,2], Chen Zhang[1], Yi Yang[6], JiaLi Wang[1], Wei Wu[7], Yi Huang[8], Yanping Tian[1], Guangxing Chen [3✉], Junlei Zhang [1✉] & Rui Jian [1✉]

A comprehensive and precise definition of the pluripotency gene regulatory network (PGRN) is crucial for clarifying the regulatory mechanisms in embryonic stem cells (ESCs). Here, after a CRISPR/Cas9-based functional genomics screen and integrative analysis with other functional genomes, transcriptomes, proteomes and epigenome data, an expanded pluripotency-associated gene set is obtained, and a new PGRN with nine sub-classes is constructed. By integrating the DNA binding, epigenetic modification, chromatin conformation, and RNA expression profiles, the PGRN is resolved to six functionally independent transcriptional modules (CORE, MYC, PAF, PRC, PCGF and TBX). Spatiotemporal transcriptomics reveal activated CORE/MYC/PAF module activity and repressed PRC/PCGF/TBX module activity in both mouse ESCs (mESCs) and pluripotent cells of early embryos. Moreover, this module activity pattern is found to be shared by human ESCs (hESCs) and cancers. Thus, our results provide novel insights into elucidating the molecular basis of ESC pluripotency.

[1] Laboratory of Stem Cell & Developmental Biology, Department of Histology and Embryology, College of Basic Medical Sciences, Army Medical University, Chongqing 400038, China. [2] Department of Pathophysiology, College of High Altitude Military Medicine, Army Medical University, Chongqing 400038, China. [3] Department of Joint Surgery, The First Affiliated Hospital, Army Medical University, Chongqing 400038, China. [4] State Key Laboratory of NBC Protection for Civilian, Beijing 102205, China. [5] Department of Cell Biology, College of Basic Medical Sciences, Army Medical University, Chongqing 400038, China. [6] Experimental Center of Basic Medicine, College of Basic Medical Sciences, Army Medical University, Chongqing 400038, China. [7] Thoracic Surgery Department, Southwest Hospital, The First Hospital Affiliated to Army Medical University, Chongqing 400038, China. [8] Biomedical Analysis Center, Army Medical University, Chongqing 400038, China. [9] These authors contributed equally: Yan Ruan, Jiaqi Wang, Meng Yu. ✉email: cgx7676@hotmail.com; zhangjunlei@aliyun.com; jianruilq2@aliyun.com

Embryonic stem cells (ESCs) have multilineage differentiation potential, and can proliferate indefinitely under defined conditions in vitro[1]. Therefore, ESCs provide an excellent model system for studying early developmental events of embryogenesis and abundant biological materials for regenerative medicine or cell therapy[2]. Understanding the molecular mechanisms of pluripotency maintenance not only promotes advances in the applications of ESCs, but also facilitates the progress of induction of pluripotent stem cell (iPSC) technology and cancer research[3].

The maintenance of pluripotency and self-renewal of ESCs require specific extrinsic signals and a hierarchical, interconnected gene network[4]. The core transcription factors (TFs) Nanog, Sox2, and Oct4 act as central units, together with other pluripotency factors, such as Myc, Esrrb, Klf4, Prdm14, Stat3, Smad and Tbx3, to form the pluripotency gene regulatory network (PGRN), which directly controls the ESC-specific gene transcription program[5]. These TFs can form large protein complexes by physically interacting with each other and associate with epigenetic factors to regulate their targets co-operatively[6]. Therefore, the ESC-specific gene regulation program is more efficiently explained by complex regulatory interactions of numerous factors in the PGRN than by the roles of individual independent genes.

Previous studies have suggested that the PGRN can be divided into sub-classes, such as the CORE, MYC and PRC classes, according to co-occupancy targets of TFs and epigenetic factors[7]. These sub-classes are functionally independent and serve as hubs of the PGRN by integrating transcriptional signals to regulate the specific gene sets and function[7,8]. Comprehensive analysis and precise definition of the regulatory sub-units of the PGRN are crucial for elucidating the regulatory mechanisms of pluripotency in ESCs. To this end, several attempts have been made to construct a more detailed PGRN. These studies have extended the PGRN primarily by adding target genes that are highly expressed in ESCs or that physically interact with previously known factors[6,9–11]. However, high expression and physical interactions cannot guarantee their essentiality in regulating pluripotency. Additionally, insufficient omics data and analysis methodology also limit deciphering the gene regulatory landscape. Therefore, an understanding of the pluripotency regulation network remains largely incomplete.

Recently, the successful application of CRISPR-Cas9-based gene knockout in eukaryotic cells has provided a new option for functional genomics screening[12,13]. This technology allows direct modifications of genomic loci, showing high knockout efficiency and low off-target effects[14]. Moreover, numerous omics data available in public databases including transcriptomics, proteomics, epigenomics and chromatin conformation maps, have facilitated the delineation of protein-protein or protein-DNA interaction networks and have promoted the identification of global target genes for TFs[15–17]. These tools have enabled researchers to develop a more comprehensive understanding of transcriptional networks and to clarify the mutual cooperation and regulation mechanism between genes.

In the present study, through a CRISPR/Cas9-mediated functional genomics screen and multi-omics integrative analysis, we established a new PGRN containing six independent transcriptional modules (CORE, MYC, PAF, PRC, PCGF and TBX). Furthermore, we characterized the activity pattern and functions of the re-defined modules in early embryo development, m/hESCs and cancer cells.

## Results

### Genome-scale CRISPR screen to identify regulators that maintain mESC pluripotency.
To establish a function-based PGRN, we first performed a CRISPR-Cas9 mediated genome-wide screen to detect genes essential for self-renewal. mESCs were cultured under Leukaemia inhibitory factors (LIF)/serum condition (L/S), which was commonly used in similar tasks and confer a naïve state to pluripotency[18,19]. For a comprehensive screen, the Brie library was chosen, which can target 19,674 genes, with high coverage across the genome[20]. Cas9-expressing R1 ESCs were infected with lentiviruses containing the library. The cells were propagated in L/S culture and collected on day 0 (P.Sc_0d) and day 14 (P.Sc_14d) post-screen (Fig. 1a). We sequenced the pre-transfected plasmid library and the P.Sc_0d and P.Sc_14d cell samples. The results revealed the presence of 99.79% single-guided RNA (sgRNA) in the plasmid library and a mean of 166 reads per sgRNA (Fig. 1b). In the P.Sc_0d samples, the sgRNA presentations were 99.49% and 99.40% in two biological replicates (Fig. 1b), correlating highly with the plasmid representation (r = 0.73 on average) (Fig. 1c, d). The sgRNA representations of the P.Sc_14d samples also showed high concordance between biological replicates (Fig. 1d). The sgRNAs with significantly increased or decreased abundance were almost exclusively observed for expressed genes (RPKM > 0.5). The abundances of the sgRNAs targeting non-/low- expressed genes (RPKM ≤ 0.5) remained the same as the initial pool (P.Sc_0d) (Fig. 1e).

Using MAGeCK[21], we detected 2930 genes whose sgRNAs were depleted, suggesting those as genes required for mESC fitness, as well as 1384 genes whose sgRNAs were enriched, indicating genes harmful to the self-renewal of mESCs (Supplementary Data 1). Despite statistical differences, the positive selection genes showed low fold-change values (only 14 genes with log2-Fold Change ≥1) and were mostly related to growth restriction and lineage development. Thus, we focused on the negative selection genes that were essential for self-renewal maintenance of mESCs. These genes were distributed across all chromosomes without enrichment in specific chromosomal regions (Supplementary Fig. 1a). A total of 44.3% of the identified genes encoding proteins localized in the nucleus, 35.2% were located in mitochondria, and the rest were distributed in the cytosol, and among ribosomes and the cytoskeleton (Supplementary Fig. 1b). Twenty-five percent of the genes encoded nucleic acid-binding proteins, TFs and chromosome-associated proteins, while the rest were metabolic enzymes and plasma membrane proteins (Supplementary Fig. 1c). GO analysis of these genes showed that the enrichments were associated with fundamental cellular processes for cell survival (Fig. 1f, Supplementary Fig. 1e, f and Supplementary Data 1). Correspondingly, the majority of the genes involved in RNA transport, ribosomes and DNA replication were identified and included in this group (Fig. 1g).

To examine the correlation between the negative selection genes and pluripotency, we used the low-expression[22] and non-essential genes[23] as negative controls, the ribosome genes and core TFs in ESCs as the positive controls, and compared the target gene sgRNA abundance in the P.Sc_14d and P.Sc_0d samples. The results showed that the sgRNAs targeting the ribosome and core TFs genes were significantly decreased, whereas little change was observed in those targeting the low-expression and non-essential genes (Fig. 1h). When compared to the non-essential genes, the top100 and top1000 targeted genes ranked by MAGeCK had significantly higher expression levels in mESCs (Fig. 1i). Enrichment analysis also indicated that the negative selection genes were enriched in biological processes and pathways involved in ESC self-renewal (Supplementary Fig. 1d, g). The highest-ranking genes in the screen included the core factors Oct4, Nanog and Sox2, and several genes not previously implicated in ESC self-renewal maintenance, such as Zpr1, Ykt6 and Zbtb17 (Fig. 1j). These genes were all chosen as candidates for constructing the PGRN.

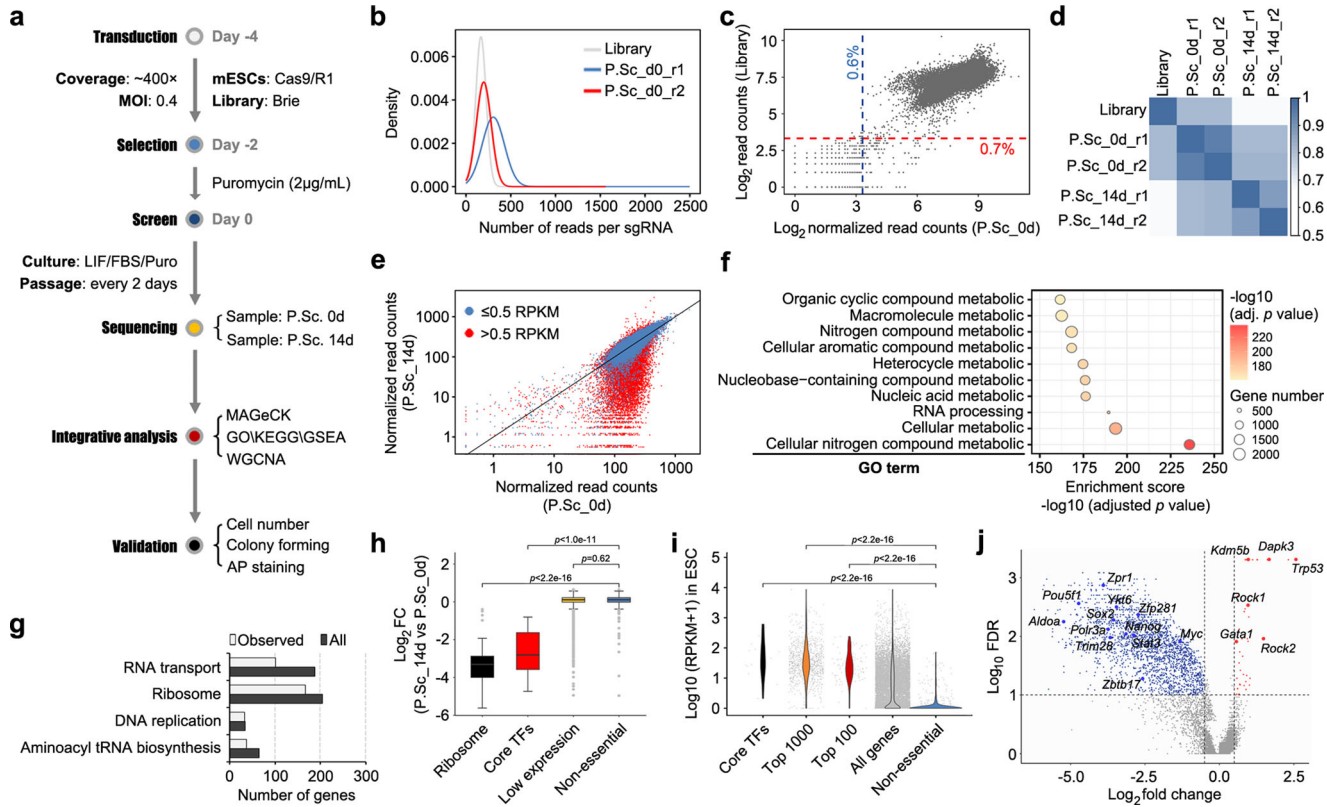

**Fig. 1 CRISPR/Cas9-based knockout screen in mESC pluripotency maintenance. a** Schematic of loss-of-function screening with the Brie library.
**b** Distributions of the number of reads per sgRNA in the library (grey), and the P.Sc_d0_r1 (blue) and P.Sc_d0_r2 (red) samples. **c** Scatter plots comparing
the sgRNA read counts in the plasmid library and P.Sc_d0 samples. Red and blue lines indicate that 0.7% and 0.6% of the sgRNAs have undetectable
representations (less than 10 reads). **d** Heatmap of Pearson correlation coefficients among the library and the P.Sc_d0 (with two biological replicates r1 and
r2) and P.Sc_d14 (with two biological replicates r1 and r2) samples. **e** Comparisons of gene read counts between the P.Sc_d0 and P.Sc_d14 samples. The
red dots represent expressed genes (>0.5 reads per kilobase per million mapped reads (RPKM)), and the blue dots represent low-/non-expressed genes
(≤0.5 RPKM) in mESCs. **f** Biological processes enriched in the negative selection genes. Top 10 enrichment terms are presented. **g** The column chart
indicates the numbers of genes involved in the fundamental cellular processes. Observed, genes identified in negative selection. All, all the genes involved
in the cellular processes (KEGG database). **h** The fold changes of the sgRNAs targeting the "ribosome", "core TFs", "low expression" and "non-essential"
genes (Supplementary Data 4) in the P.Sc_d14 samples relative to the P.Sc_d0 samples. The data are represented as $\log_2$ FC (fold change of sgRNA read
count). $p$ values were calculated using the Wilcoxon signed-rank test. **i** Expression levels of the "core TFs", "Top 1000" (top 1000 genes ranked by
MAGeCK), "Top 100" (top 100 genes ranked by MAGeCK), "all genes" and "non-essential" genes in mESCs (Supplementary Data 4). The values are
represented as $\log_{10}(RPKM + 1)$. $p$ values were calculated using the Wilcoxon signed-rank test. **j** A volcano plot of the screen results. The blue/red dots
indicate the negative/positive selection genes with an absolute sgRNA $\log_2$ fold change >0.5 and FDR < 10%.

**Generation of an extended self-renewal gene set by integrating
different screening data**. Since results from a single screen might
be influenced by the specific CRISPR library used or by other
factors, we then compared our list of negative selection genes to
those identified in four previous screens performed under the
same L/S culture conditions but with different mESC lines and
CRISPR libraries[13,24–26]. Unexpectedly, there was only 1 common
gene between the five screens (Supplementary Fig. 2a). Con-
sidering that the different analysis methodologies used may
influence the readout, we re-analysed the raw data of these
screens with MAGeCK and identified genes that were sig-
nificantly changed ($p < 0.05$) under negative selection. Pearson
correlation analysis of these normalized data showed high con-
cordance between the Tzelepis and Li screens and our screen.
However, the screens by Zhao and Shohat pointed to a unique set
of genes (Fig. 2a). In total, 457 (11.26%) genes were identified in
all five screens (defined as the "common" gene set), while 3601
(88.74%) genes were identified in four or fewer screens (defined
as the "context-specific" gene set) (Fig. 2b, Supplementary
Fig. 2b). To test whether the context-specific genes were false
positives caused by different screens, we assessed the functional

relevance of these genes in self-renewal by examining their
expression levels in E14 ESCs using RNA-seq data from GEO[22].
Comparatively, despite being lower than the core TFs gene set,
both the common and context-specific genes showed significantly
higher expression levels than the non-essential genes (Fig. 2c). To
exclude any bias caused by cell lines and culture conditions, we
analysed the single-cell RNA-sequencing (scRNA-seq) data of
IB10 ESCs[27] and inner cell mass (ICM) cells from E4.5d
embryos[28]. Indeed, we observed high concordance of expression
between the common genes, context-specific genes, core TFs and
highly expressed genes in mESCs (Supplementary Fig. 2c, d).
To further clarify whether the high expression of the common
and context-specific genes was specific to self-renewing mESC,
weighed gene co-expression network analysis (WGCNA) was
performed to analyse the gene expression profiles during mESC
differentiation. As shown in Fig. 2d, six co-expression modules
were constructed. Module-trait relationship analysis indicated
that the blue module, which contains 86.35% of the common and
72.43% of the context-specific genes, had a high correlation with
mESCs (Fig. 2e) and was significantly downregulated during
either embryonic body (EB) formation or directional

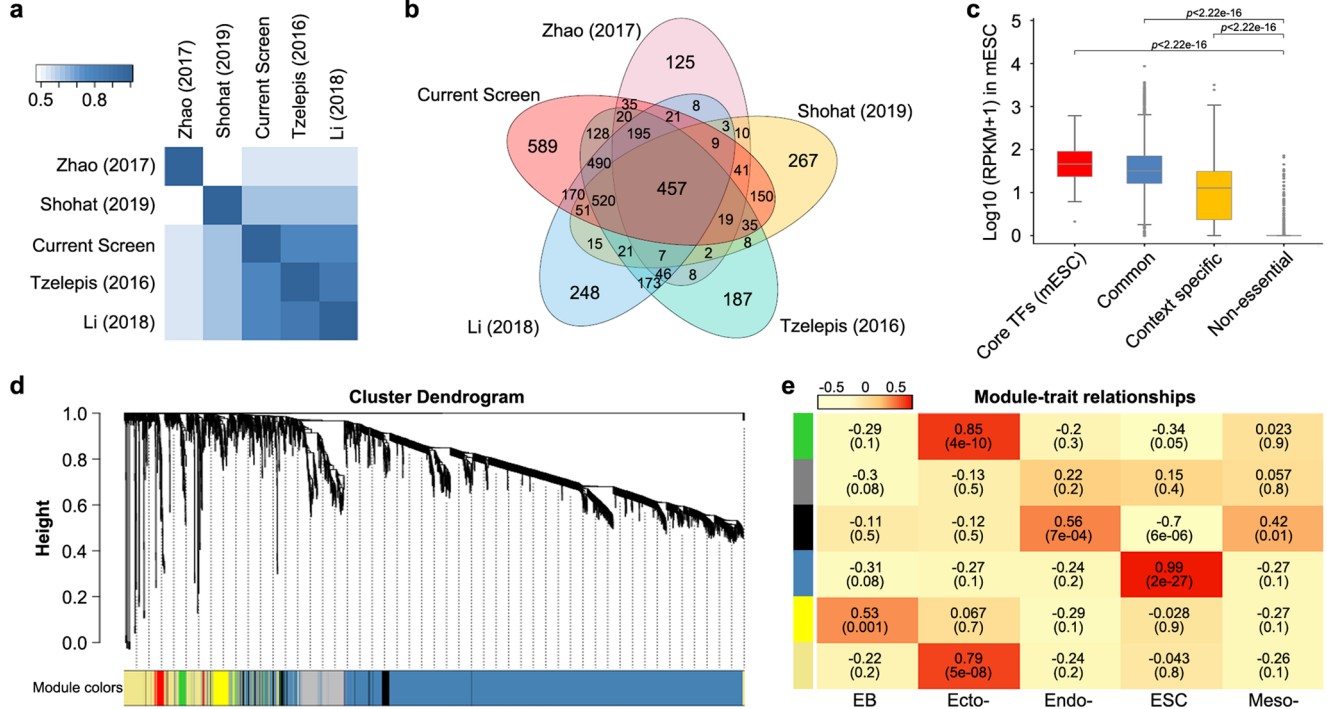

**Fig. 2 Comparing and integrating the data of other screens. a** Heatmap of Pearson correlation coefficients among all pairs of screens. **b** Venn diagram depicting the comparison of different screening studies with normalized data. The 457 overlapping genes were defined as the common set, whereas all the other genes were defined as the context-specific set (Supplementary Data 5). **c** Expression levels of the "core TFs", "common", "context-specific" and "non-essential genes" in mESCs. The data are represented as $\log_{10}$ (RPKM + 1). **d** Cluster dendrogram and module assignment for modules from WGCNA. Common and context-specific genes were clustered based on a topological overlap matrix (TOM). The branches correspond to modules of highly interconnected groups of genes. Colours in the horizontal bar represent the modules. Six modules with 3303 genes were detected. **e** Heatmap of the relationships between the modules and samples. Colours in the longitudinal bar represent the modules. The red cells correspond to positive correlation. The yellow cells correspond to negative correlation. The numbers in the cell indicate the correlation coefficient (upper) and *p*-value (below). Colour intensity is proportional to the correlation coefficient.

differentiation (Supplementary Fig. 2e), suggesting a strong functional relevance in mESC self-renewal.

**Functional validation of the candidate genes in maintaining mESC self-renewal.** To prove the veracity of the screening and integrative analysis, four genes (*Ykt6, Polr3a, Adoa, Wdr75*) in the common set and six genes (*Serbp1, Zbtb17, Zpr1, Usp8, Pi4a* and *Bap1*) in the context-specific set were selected to further validate their functions in mESC self-renewal. For each candidate, two sgRNAs were designed, and two mESC lines, R1 and CCE, were used to assess the self-renewal phenotype. Because both sgRNAs behaved similarly in two cell lines, data is presented for only one. The silencing level of each sgRNA was measured and confirmed by qRT-PCR (Supplementary Fig. 3). Compared to the wild-type (WT) and control (non-targeting sgRNA transduction) cells, all the target gene knockout (KO) cells, except for the Ykt6 KO cells, showed significantly reduced proliferation and colony forming capacity (Fig. 3a, b). Morphological observation and alkaline phosphatase (AP) staining assays showed that all 10 target gene KO cells displayed a differentiation-prone phenotype, i.e., flattened colony morphology, more scattered differentiation-like cells, and fewer AP-positive colonies (Fig. 3c, d). Collectively, these results suggest that both the common and context-specific genes are valid hits and required for the maintenance of self-renewal in mESCs.

**Characterization of the integrated self-renewal related gene set.** We next combined the core and context-specific genes and defined them as the integrated self-renewal related gene set

(iSRGS) (Supplementary Data 6). The enrichment analysis showed that the iSRGS genes were mainly associated with fundamental cellular pathways such as DNA replication, proteasome degradation, oxidative phosphorylation, and the cell cycle (Fig. 4a, Supplementary Data 6). To ascertain the cellular pathways specific to mESCs, we performed gene set enrichment analysis (GSEA) to analyses the expression profiles of genes in enriched pathways. The results showed that genes involved in the "oxidative phosphorylation", "ubiquitin proteasome", "mRNA processing" and "translations" pathways were highly expressed in mESCs and downregulated after differentiation (Fig. 4b). Since these pathways are common across cell types[29,30], these results suggest a possibly specific functional gene set of these cellular pathways in mESCs.

Signal pathways involved in the pluripotent state, such as the Wnt, JAK/Stat, TGFβ, p53 and FGF pathways[18], were also significantly enriched. Moreover, our results showed enrichment of pathways that were not reported to be involved in pluripotency regulation, such as the androgen receptor signalling pathway and Epo signalling pathway. In addition, pathways involved in the immune system, including T/B cell receptor signalling and IL-7/IL-6 signalling pathways, were also enriched (Fig. 4a).

**Reconstruction of the PGRN in mESCs.** To re-construct the PGRN, we incorporated the transcriptional regulators in the iSRGS into the known regulatory networks, and clustered the regulatory units according to co-occupancy targets using ChIP-seq datasets that were available in public databases. As a result, we obtained nine sub-classes (Fig. 5a). In contrast to previous

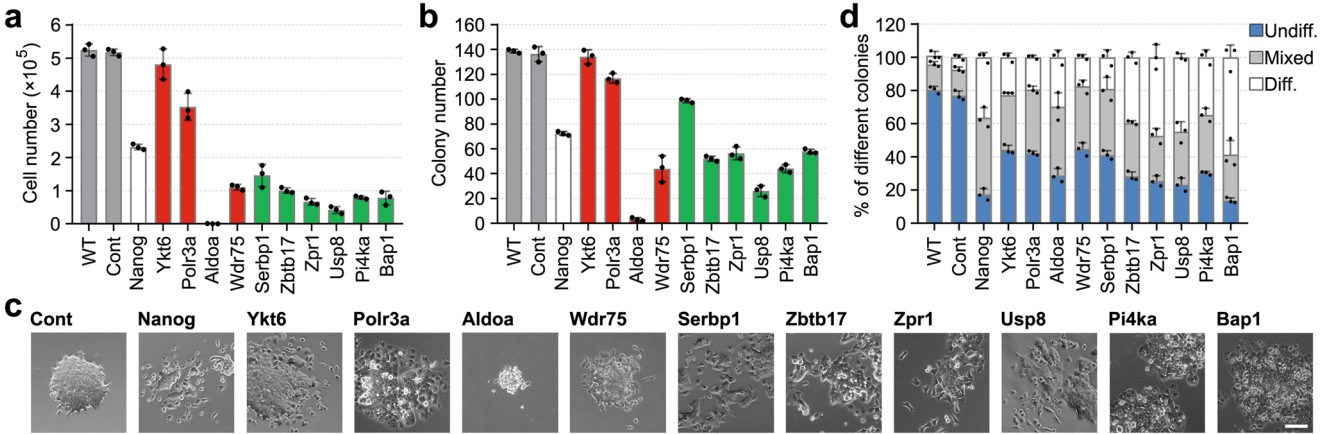

**Fig. 3 Phenotypic validation of the candidate gene. a** The proliferation rates of WT, Cont (cells infected with control sgRNA) and target gene knockout cells. The indicated cells (1000 cells per cm² in 12-well plates) were cultured in L/S for four days, and the cell numbers were counted. **b** Colony numbers of the indicated cells. Cells were seeded into 12-well plates at a density of 200 cells per well and grown for 6 days, and colony numbers were counted. **c** Morphology of colonies formed by the indicated cells. Scale bar, 200 μm. **d** Colonies were fixed and stained for AP and then scored as undifferentiated, mixed, or differentiated. Data in (**a**, **b** and **d**) are represented as the mean ± SD; n = 3.

reports, the contents of the CORE, PRC and MYC sub-classes were all significantly increased, whereas regulators in the CTCF, REST and P53 classes were almost unchanged[31]. In addition, three new classes, PCGF, PAF and TBX, were identified and named based on the representative factors in each class. The PCGF class consisted of PcG protein-related transcriptional repressors and methyltransferases. The PAF class included the PAF1 complex, H3K9me3 binding protein and mRNA methyltransferases. The TBX class included TFs of the POU and T-box family, chromatin looping proteins, nucleosome re-modelling proteins, histone-related proteins, RNA-related proteins and DNA methyltransferase (Fig. 5a, Supplementary Data 7).

The peak annotation analysis showed that the majority of binding peaks generated by factors in the MYC and PRC classes were more centred at the transcriptional start site (TSS), whereas the CORE, CTCF, P53 and REST classes generally localized further away from the TSS (Supplementary Fig. 4a). The peaks of the newly identified PAF class were located on both promoters and gene bodies, while the PCGF and TBX classes were localized mainly on introns and distal intergenic regions, which may suggest different regulatory modes (Supplementary Fig. 4a).

Since the target site of TFs is often related to specific chromatin marks[32], we then examined the association between factor co-occupancy and the histone modification signatures in each class. As shown in Fig. 5b, the CORE class binding regions were highly enriched with putative enhancer histone signature H3K4me1 and active histone signature H3K27ac. The factors in the PRC class harbored both repressive (H3K27me3) and active (H3K4me3) histone signatures on their binding sites, which indicated bivalent chromatin[33]. Targets occupied by the MYC class showed high levels of the active histone signatures H3K4me3 and H3K27ac. The occupancies of the P53 and REST classes were associated with H3K27ac and H3K4me2, respectively. The PAF class targets were enriched with the active histone markers H3K4me3, H3K79me2 and H3K27ac, while PCGF and TBX occupied targets specifically associated with the repression histone markers H3K9me3 and K4K20me3 (Fig. 5b, Supplementary Fig. 4b). These results suggested that the CORE, P53, REST, PAF and MYC classes were mainly involved in the regulation of transcriptional activation. Among them, the genes in the CORE, P53 and REST classes generally bound distal regulatory elements, whereas those in the PAF and MYC classes preferentially bound to proximal regulatory elements. The PCGF and TBX classes

typically correlated with transcriptional repression, by targeting distal silencers. The PRC class factors were enriched on poised genes by occupying proximal promoter regions with bivalent modifications.

**Establishment of individual functional modules based on the newly defined transcriptional sub-classes.** In a sub-class, TFs and their co-occupied target genes compose a regulatory module, which represents a co-operative function of factors in the sub-class. *Nearest gene* linkage is the commonly used method for calling target genes. For each binding site identified by ChIP-seq, this approach usually assigns the nearest gene as its potential transcriptional target[6,7]. Because some transcriptional sub-classes (CORE, CTCF and TBX) were preferentially located in intergenic regions >10 kb from the TSS of annotated genes (Supplementary Fig. 4a), we used adaptive sampling and an ensemble model (AdaEnsemble) to assign target genes. This approach integrated gene expression profiles with TF binding profiles and chromatin conformation data to predict high-confidence target genes regulated by both proximal and distal sites[34,35]. Accordingly, putative proximal and distal target genes were identified for six major classes (CORE, PRC, MYC, PAF, PCGF and TBX). The other classes were not investigated further, as they consisted of non-specific chromatin looping regulators (CTCF) or just a single factor (REST and P53). As shown in Fig. 6a, while the PCGF module contained a similar ratio of proximal and distal targets, the MYC and PRC modules had more putative proximal target genes (59.5% and 67.6% respectively), and the CORE, PAF and TBX modules had more putative distal target genes (69.4%, 60.1% and 58.2% respectively). These results were in line with the distribution characteristics of binding sites of the individual sub-classes (Supplementary Fig. 4a).

Lists of the module gene sets were summarized in Supplementary Data 9. The PRC, PCGF and TBX modules showed clear separation, whereas the PAF, MYC and CORE modules shared many targets with each other (Fig. 6b, c). It was shown that there were 143 intersecting genes between the PAF and MYC modules and 106 intersecting genes between the PAF and CORE modules (Fig. 6c). As the CORE, PAF and MYC classes were mainly involved in the regulation of transcriptional activation, these results indicate that the promoters of active genes may always be bound by multiple factors, whereas repressed genes were regulated by fewer factors. To test whether the modules were

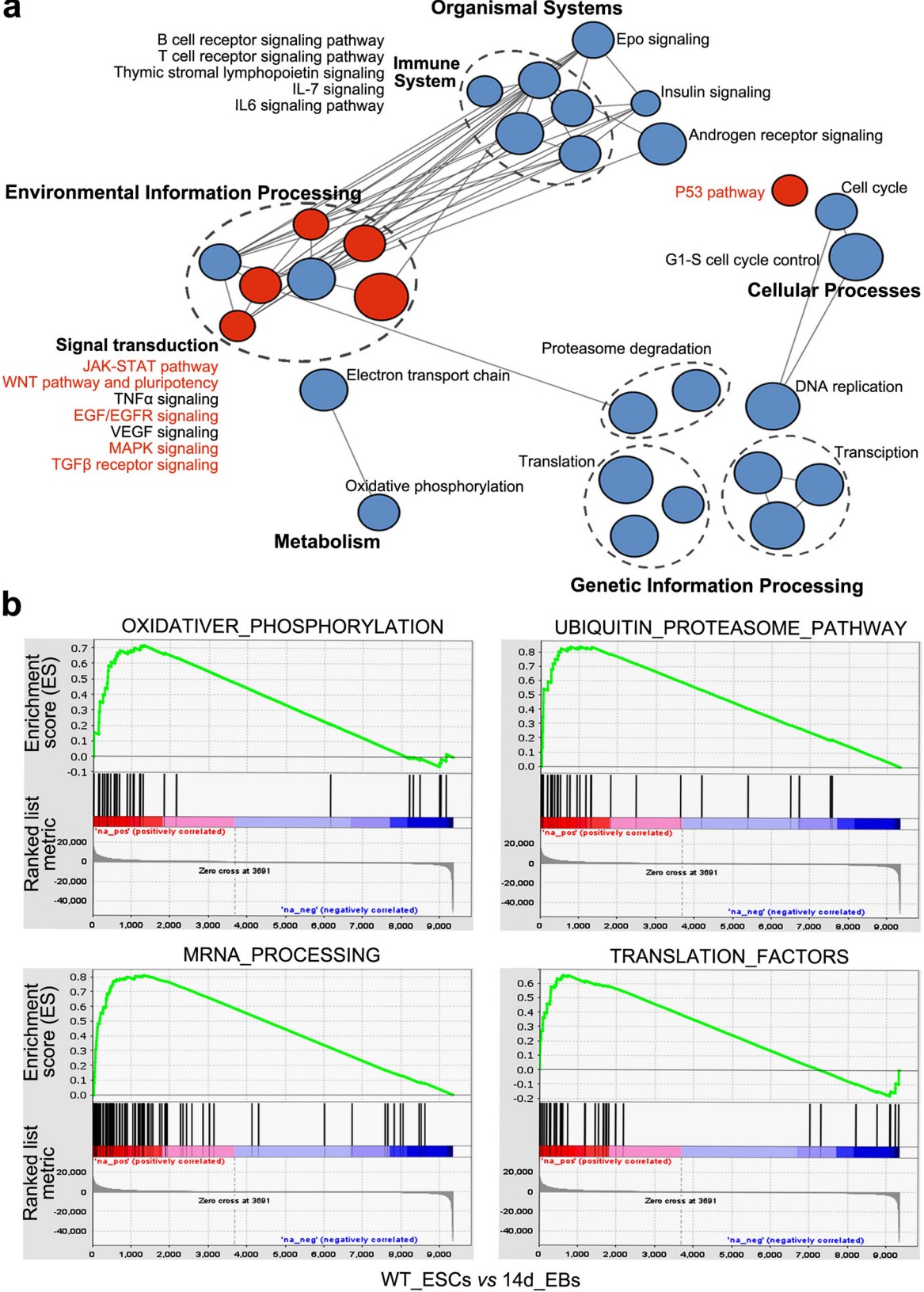

**Fig. 4 Enrichment analysis of the iSRGS. a** Enrichment map networks of pathway terms enriched by the iSRGS. The red nodes indicate pathways that were reported to participate in pluripotency maintenance. **b** GSEA showed the expression of fundamental pathway genes in ESCs relative to EBs differentiated for 14 days.

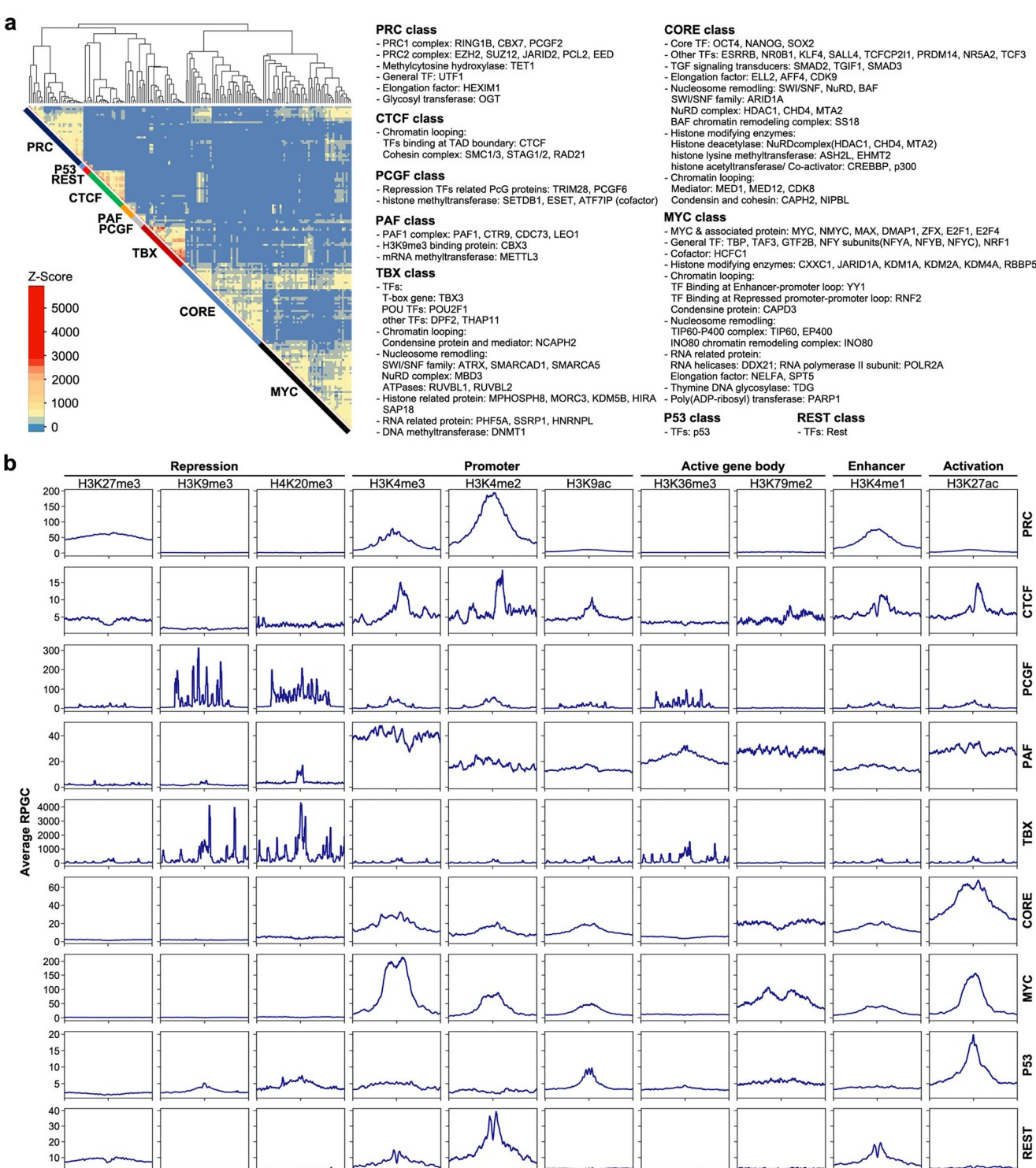

**Fig. 5 Clustering the sub-classes in the PGRN and identifying their associated histone modification status. a** Classifications of sub-classes based on co-occupancy. Nine sub-classes were unsupervised hierarchical clusters based on Z-scores from 171 genes (total of 374 ChIP-seq data, see Methods section). All the sub-classes and their contents are shown on the right: PRC (12 genes, 29 experiments), CTCF (7 genes, 17 experiments), PCGF (5 genes, 5 experiments), PAF (6 genes, 6 experiments), TBX (20 genes, 21 experiments), CORE (32 genes, 41 experiments), MYC (43 genes, 47 experiments), P53 (single gene, 2 experiments) and REST (single gene, 3 experiments). **b** Histone modification status of the nine sub-classes. The average normalized read count of histone modification (y axis) within ±3 kb from the central peak of the sub-class (x axis) is plotted.

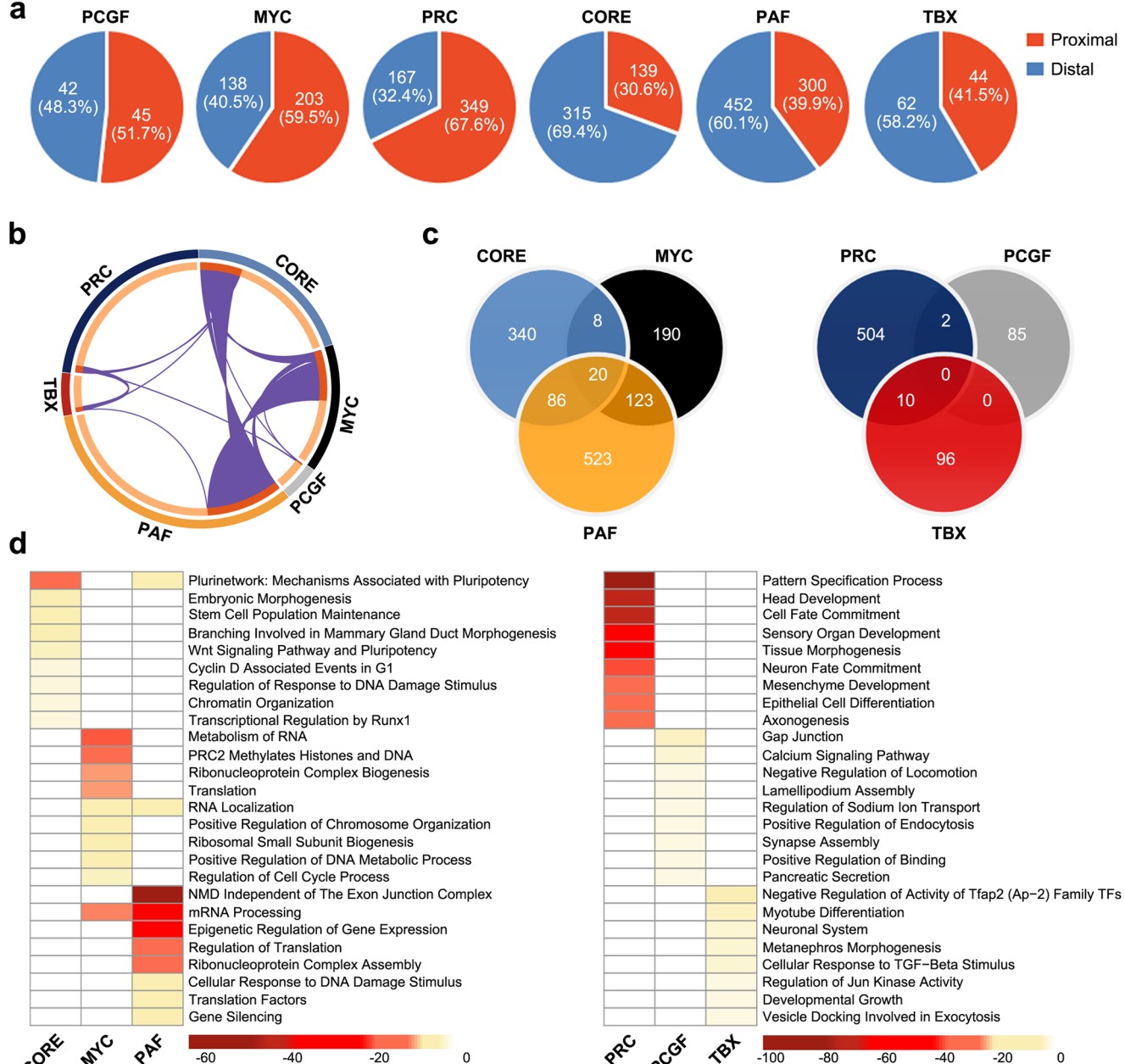

**Fig. 6 Gene composition and functional enrichment analysis of each module. a** The percentage of proximal and distal target genes defined by AdaEnsemble (see Method section) in each module. **b** Circos plot indicates the overlaps between module genes. The inner circle represents the gene lists of each module. Genes that hit multiple lists are coloured in dark orange, and genes unique to a list are shown in light orange. The purple curves inside link identical genes. **c** Venn diagram demonstrating the gene numbers and overlaps in transcriptionally active modules (left) and transcriptionally repressed modules (right). **d** GO analyses of genes in transcriptionally active modules (left) and transcriptionally repressed modules (right) (Supplementary Data 10). The top10 enrichment terms in each module are shown. The cell colour corresponds to –log$_2$ (p-value).

functionally separable, we performed GO analysis and found that each module was primarily involved in different biological processes (Fig. 6d), suggesting that the modules are functionally independent.

Since the same CORE, MYC and PRC modules were defined in a previous report[6], we examined the similarity of two gene sets in each module. Comparatively, 454 CORE, 341 MYC and 516 PRC genes were identified in this study, whereas 111, 503 and 560 corresponding genes were identified by Kim et al. (hereafter referred to as CORE-Kim, MYC-Kim and PRC-Kim). Nonetheless, only 40, 62 and 233 genes overlapped between the respective modules (Supplementary Fig. 5a). The functional

enrichment analysis also showed that different GO terms and pathways were identified in each respective module (Supplementary Fig. 5b–d). Some known pluripotency-associated pathways, such as the Wnt, PI3K/Akt and Hippo/Yap pathways, were specifically enriched in the current CORE module. In contrast, in the CORE-Kim module, more development-related pathways were enriched (Supplementary Fig. 5b). These results indicate remarkably different gene compositions and module functions between the previous and current studies.

For validating the newly defined PAF, PCGF and TBX3 modules, the ChIP assay have been performed using anti-Flag antibody in Flag-tagged Ctr9 (for PAF module), Tbx3 (for TBX

module), Dpf2 (for TBX module) and Pcgf6 (for PCGF module) transfected mESCs. Then 11 target genes in different modules were chosen for qPCR analysis. The results were consistent with previous reports[36–38] and showed that all the four transcriptional factors preferentially occupied the targets of their own modules (Supplementary Fig. 6), indicating the specific and reliable assignment of target genes for each module. Furthermore, functional analyses were performed to assess the impact of the newly constructed modules on mESCs self-renewal. The major transcriptional factor of each module (Ctr9, Pcgf6 and Tbx3) was knocked down individually by RNAi in a Nanog-GFP reporter ESC line[39] (Supplementary Fig. 7a). Compared to the WT ESCs, Ctr9, Pcgf6 and Tbx3 KD cells displayed an impaired self-renewal phenotype characterized by slower proliferation rate, decreased proportion of AP-positive colonies and fewer Nanog-GFP positive cells (Supplementary Fig. 7b–d). These results indicated that PAF, PCGF and TBX3 modules were essential for mESC self-renewal. Moreover, we compared the RNA-seq data from mESCs in which the major component of CORE, MYC, PAF, PRC, PCGF and TBX3 module were silenced respectively. The GO analysis for DEGs in each gene KD cells showed that distinct prominent terms were enriched, despite all of them were involved in development regulation (Supplementary Fig. 7e), which further indicated that different module may have separate functions in mESC self-renewal maintenance.

**Mapping the module activity patterns during mESC differentiation and embryo development**. Next, we tested the module activities in mESCs and differentiated cells. GSEA revealed that the genes in the CORE, MYC, and PAF modules were highly expressed in mESCs and downregulated after differentiation. In contrast, the genes in the PRC, TBX and PCGF modules were repressed in mESCs (Fig. 7a). We additionally tested the activity of each module during directed ectoderm, endoderm and mesoderm differentiation of mESCs. As expected, the CORE, MYC and PAF modules were highly active in mESCs and became repressed after differentiation, whereas the PRC, TBX and PCGF modules showed the opposite activity pattern (Fig. 7b).

We then mapped the spatiotemporal module activity patterns during mouse embryo development from E2.5 to E7.5 based on published scRNA-seq data[40,41]. As shown in Fig. 7c, the CORE, MYC and PAF modules were highly active in pluripotent cells (MOR in E2.5, ICM in E3.5, epiblast in E4.5-E7.5) and repressed in differentiated cells (E and M in E5.5-E7.5), whereas the activities of the PCGF and TBX modules displayed almost opposite patterns. In comparison, the PRC module genes showed a similar expression pattern to those of the PCGF and TBX module genes except for the expression in the early stages of the pre-implantation embryo (MOR in E2.5 and ICM in E3.5) (Fig. 7c), suggesting a possible stage-specific function of this module in early and late embryo development. Together, these data reveal consistent correlations between the module activities and the pluripotent states of cells both in vitro and in vivo.

**Module activity in hESCs**. hESCs have a multilineage differentiation potential similar to that of mESCs[42]. In addition, core mESC TFs, such as Oct4 and Tbx3, are active in hESCs and directly participate in pluripotency maintenance[43,44]. Using the gene expression profiles of the hESCs and EB samples, we tested whether the module activity pattern was similar between mESCs and hESCs. The results showed that the activities of the six modules in hESCs were comparable to those in mESCs (Fig. 7d). To exclude cell line-specific effects, we performed analyses in both H1 and H9, and consistent results were obtained. These observations suggest conserved roles for these modules in human and mouse

ESCs. Since hESCs are usually believed to be at the primed pluripotent state[45], these results are in accordance with the data obtained from in vivo development and indicate limited variation in the module activity pattern in different pluripotent states.

**Module activity in human cancers**. According to previous reports, human tumours, especially poorly differentiated tumours display an ESC-like expression signature that may result from re-wiring of stem cell regulatory circuits[46,47]. Therefore, ESC-like gene modules have been widely used in the assessment of cancer gene signatures. To test the activity of the re-defined ESC modules and establish relevance with human cancers, we first analysed the expression profile of 750 gliomas, which included 200 low-grade gliomas (LGG; astrocytomas and oligodendrogliomas of grades 2 and 3) and 550 glioblastoma multiforme of grade 4 (GBM). As shown in Fig. 8a, we observed that the average activation levels of the CORE, MYC and PAF modules were higher in gliomas than in normal brain samples and showed a positive correlation with tumour grade. Conversely, the PRC, TBX and PCGF modules showed low activation in gliomas (Fig. 8a, Supplementary Fig. 8a). In GBMs, the same module activities as in ESCs were observed, in which the CORE, MYC and PAF modules were highly active, whereas the other modules were repressed. In LGG and normal tissues, however, these modules displayed an opposite activity pattern (Fig. 8b). These results indicate that the CORE, MYC and PAF modules may also be involved in the maintenance of the malignant phenotype of cancers.

To assess whether the activity of the modules is associated with tumour prognosis, we performed Kaplan-Meier analyses of the progression-free interval (PFI) for patients. As expected, tumours that displayed the strongest CORE, MYC or PAF module activity (top 50% of the samples) were associated with significantly worse survival outcomes than tumours with the weakest module activity. In contrast, tumours with increased activities of the TBX and PCGF modules tended to be associated with better survival outcomes, whereas the PRC module did not show significant correlations (Fig. 8c).

Next, we examined whether the module activity patterns of mESCs were present in other cancers. Analyses of the public gene expression profiling datasets of bladder, breast and non-small-cell lung cancer revealed that the high-grade tumours displayed high activities of the CORE, MYC and PAF modules with repressed expression of the the PRC, TBX and PCGF modules (Supplementary Fig. 8b–f). Together, these results suggest that ESC-specific signatures are shared by various human cancers. Nonetheless, we also observed inconsistent activity of the PRC module in lung squamous cell carcinoma (LUSC) and lung adenocarcinoma (LUAD) (Supplementary Fig. 8b), indicating underlying impacts from tumour origin or cell heterogeneity.

Furthermore, for assessing the function of CORE, MYC and PAF modules in cancer cells, Nanog (for CORE module), c-Myc (for MYC module) and Ctr9 (for PAF module) were knocked down in glioma cell line U87 respectively (Fig. 9a). As expected, all of the three gene KD cells displayed a decreased proliferation rate alongside limited colony formation capacity when compared to the control cells (Fig. 9b–d). Consistently, the same phenotypes were observed in the respective gene KD lung adenocarcinoma A549 cells (Fig. 9). These findings supported previous analysis results on clinical tumor samples (Fig. 8), demonstrating the importance of the CORE, MYC, and PAF modules in cancer progression.

## Discussion

Previously, construction of the PGRN was based mainly on individual functional studies and physical interactions between genes[48]. Recently, the application of high-throughput genetic

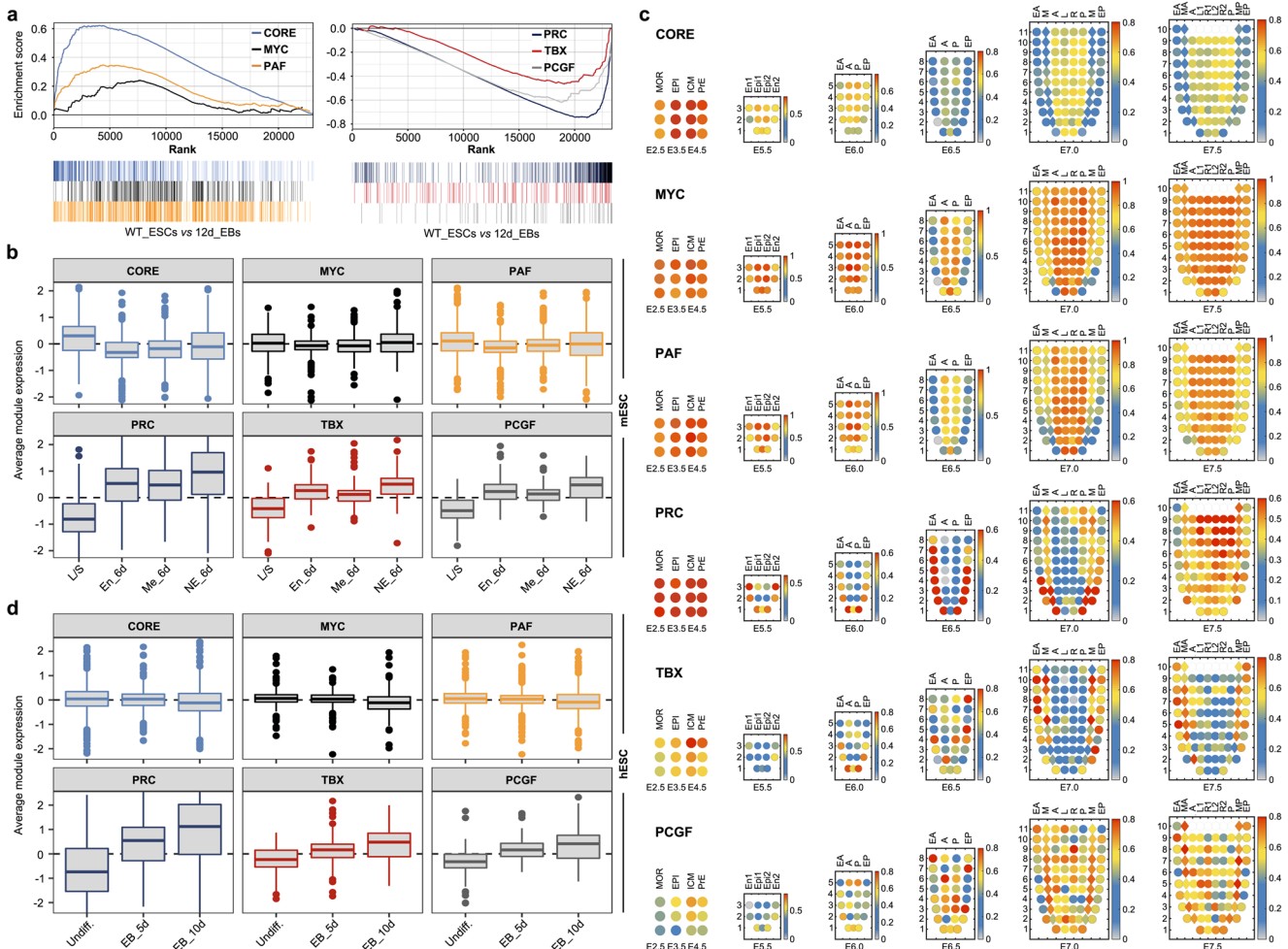

**Fig. 7 Module activity in mouse and human ESCs and early mouse embryos. a** GSEA showed the activities of the CORE, MYC, and PAF modules (upper) and the PRC, TBX, and PCGF modules (lower) in mESCs relative to 12-day EBs. **b** The boxplots show the gene expression of each module during directional differentiation of mESCs (endoderm_6 days, mesoderm_6 days, and neuroectoderm_6 days). Boxes show interquartile range, whiskers show fixed multiples of interquartile range and center line shows average gene expression values (log$_2$) (see the Methods section). **c** Mapping the module activity patterns during the mouse embryo development from E2.5 to E7.5 based on spatiotemporal transcriptomic data. The colour of the plot denotes module activity from low (blue) to high (red). **d** The boxplots show the gene expression of each module during hESC (WA09) differentiation (hESCs, EB_5 days and EB_10 days).

technologies has expanded our knowledge of pluripotency regulators. In contrast to RNAi, CRISPR-Cas9 permits more convenient, efficient and precise gene editing[14,23]. Thus, in this study, we first performed a CRISPR-Cas9-mediated functional genomics screen to systematically identify the essential genes for mESC self-renewal maintenance. Based on these results, we conducted a series of integrative analyses with another multiple-omics dataset to re-define the PGRN in mESCs.

Several studies have performed CRISPR-Cas9 functional genetic screens and obtained essential genes that are closely related to the maintenance of ESC pluripotency[13,24–26]. However, the results between screenings show significant variances, which may be due to the different sgRNA libraries, cell models or analysis methodologies used in the different studies (Supplementary Table 1). While previous studies often focused on consensus target genes to rule out false-positive hits[26], some authentic pluripotency genes may be missed. In this research, comparative analysis of five CRISPR screens performed under similar culture conditions showed that the consensus genes (common) accounted for only 10% of the target genes, whereas 90% of the target genes were context-specific. However, through expression profile analysis and functional validation, both the

common and context-specific genes were found to be necessary for the maintenance of ESC fitness. Therefore, data from different screens were integrated for further analysis to obtain more comprehensive functional genomics information. Consequently, three new transcriptional regulatory units (PCGF, PAF and TBX) were established that complemented the previously constructed sub-networks in mESCs.

The transcription program is controlled by TFs and their co-factors[5]. The co-factors lack DNA binding capacities and are recruited by TFs to regulate gene expression. Generally, co-factors are necessary for the activation/repression of transcription but are not pluripotent stem cell specific. Therefore, previous studies have focused mainly on TFs when analysing transcriptional networks or sub-classes. However, functional genomics screens in ESCs have revealed that some co-factors, such asTIP60-p400[49], PAF1C[50], Mediator and Cohesin[51], are essential for pluripotency maintenance. On the other hand, the results of ChIP-seq and liquid chromatography tandem mass spectrometry (LC-MS/MS) analyses have indicated that the collaboration between TFs and co-factors is selective. For example, in ESCs, Myc preferentially constructs a regulatory complex with co-factor Tip60-Ep400[7], whereas Oct4 mainly co-operates with Mediator and Cohesin[52].

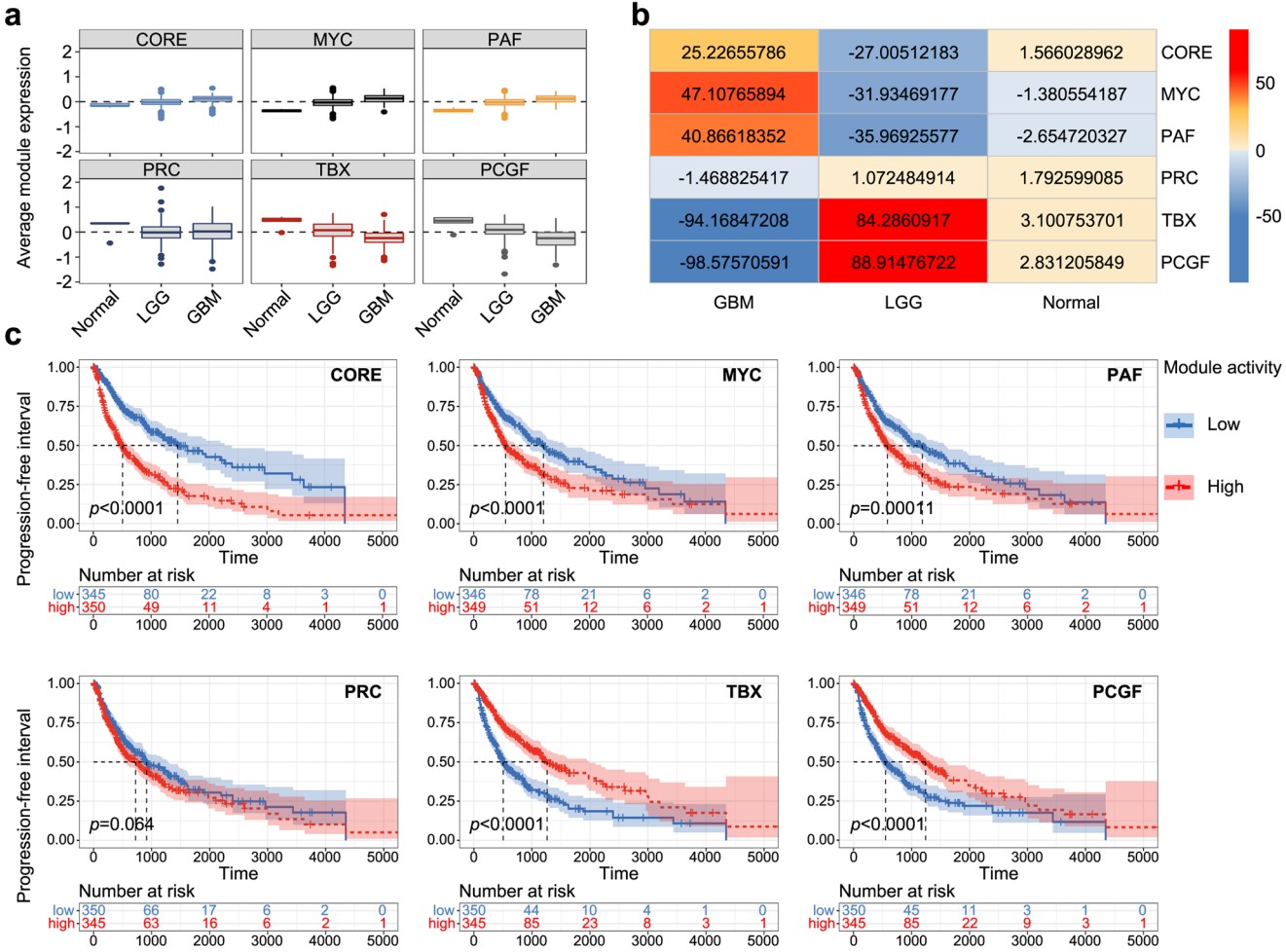

**Fig. 8 Association of module activity with tumour subtype and prognostic outcome in gliomas. a** The boxplots show the gene expression of each module in normal, LGG and GBM samples. Boxes show interquartile range, whiskers show fixed multiples of interquartile range and center line shows average gene expression values (log₂). LGG, low-grade gliomas. GBM, glioblastoma multiform. **b** Enrichment patterns of modules in normal, LGG and GBM group samples. Numbers indicate the enrichment significance (-log2 (p-value)) of module genes within each sample group. Red (positive value) indicates enrichment for high-expression, and blue (negative value) indicates enrichment for low-expression. Colour intensity corresponds to the -log2 (p-value). **c** Kaplan–Meier analyses of the progression-free interval in patients. Patients with high module activities (50% cut-off) are labelled red, whereas patients showing low module activities (50% cut-off) are labelled blue. p values are calculated using the log-rank test.

In this study, through efficient functional screening and integrative analysis, we identified a broader range of co-factors indispensable for maintaining ESC fitness, including factors involved in nucleosome re-modelling (SWI/SNF family, NuRD complex), chromatin looping (RNF2, NCAPH2, MED12, TIP60-P400 complex) and histone modification (MSL complex, CXXC1, JARID1A, KDM1A, KDM4A). These findings allowed us to define the sub-classes of the PRGN more exactly.

Identifying the downstream target gene sets of regulators determines the quality of the constructed functional module. Previous studies often used the ad hoc approach to identify target genes, such as assigns TF binding sites to the nearest genes or to genes within arbitrary genomic distance thresholds (for example, 8 kb upstream and 2 kb downstream of a TSS)[6,7,53]. However, it becomes unreliable for the assignment of genes to distant regulatory elements[35]. Since several of the sub-classes we defined in this study (CORE, PAF and TBX) showed preferential binding of distal sites, it is essential to identify distal target genes of sub-class factors. Moreover, as physical contact does not always imply functional regulation by TFs, it is necessary to distinguish functionally relevant target genes. In this study, we used AdaEnsemble to identify target genes, which predicted both proximal and distal

target genes by combining ChIP-seq and Hi-C data[35]. Furthermore, functionally relevant candidates were discriminated via expression profile analysis. Using these data and approaches, a new PRGN that comprises more detailed functional modules was defined.

The gene sets in the CORE, MYC and PRC modules were significantly different from those defined in previous studies[7]. This discrepancy could be explained by several reasons. First, compared to previously defined clusters, the CORE, MYC and PRC classes were re-established with increased TFs and co-factors. The occupancy sites in each class were changed correspondingly. Second, different criteria were used to identify the co-occupancy target genes. For instance, in Kim's research, the CORE module targets were identified as genes co-occupied by 7 of 9 factors, but in this study, the CORE targets were identified as genes co-occupied by 25 of 32 factors. Actually, upon the decrease in co-occupancy (e.g., 20 of 32 factors in CORE, 25 of 42 factors in MYC or 7 of 12 factors in PRC), the re-defined modules will cover nearly all the target genes of previously defined modules, but not vice versa. Third, different approaches were used to assign target genes. While previously defined modules focused only on the proximal targets, both proximal and distal target genes were

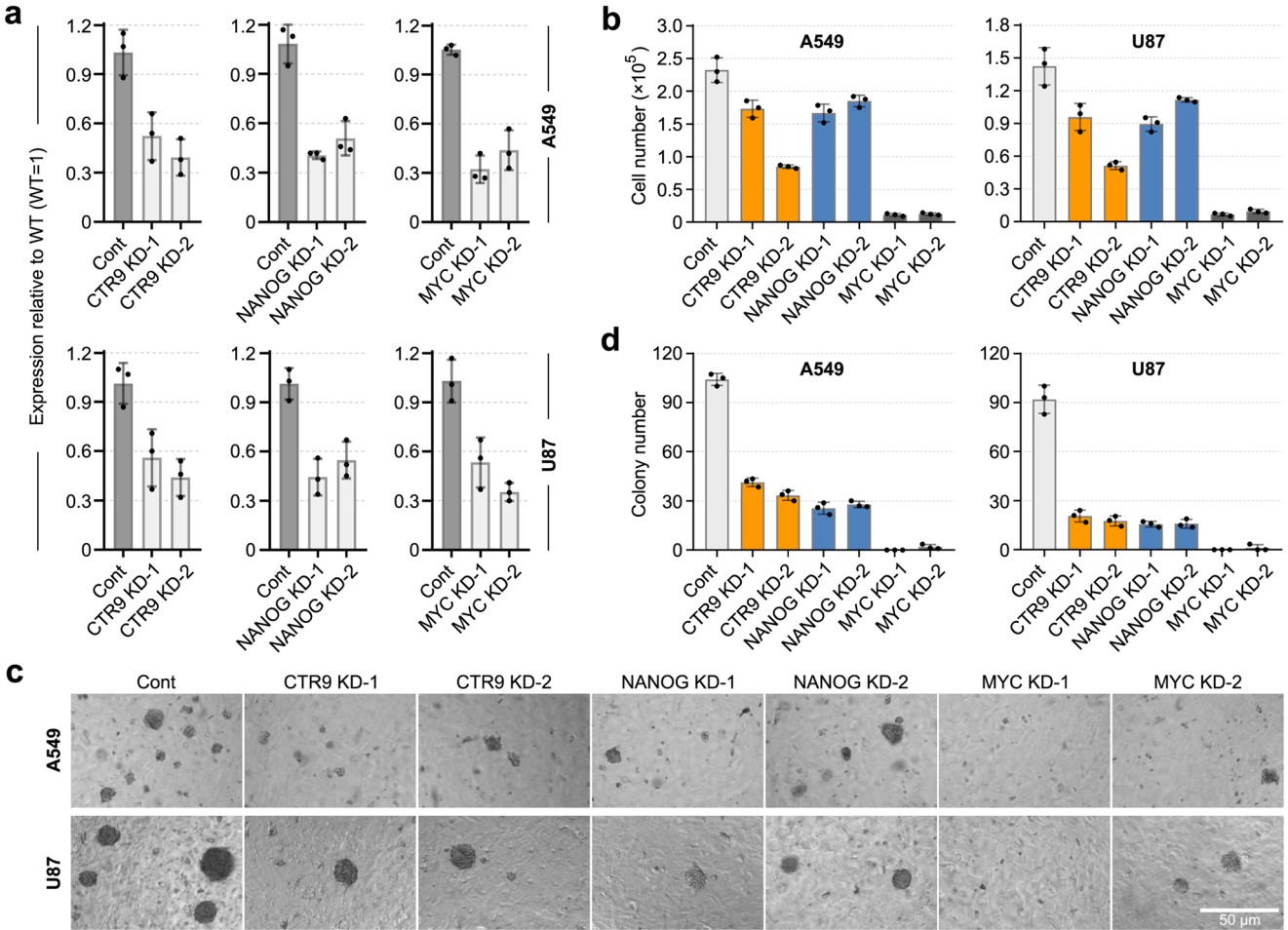

**Fig. 9 The functional analysis of CORE, MYC and PAF modules in cancer cells. a** qRT-PCR analysis for the knocking down of indicated gene expression in A549 and U87 cells. The major transcriptional factors of each module, Ctr9 (PAF), Nanog (CORE) and Myc (MYC), were chosen for functional analysis. Cells were infected with lentivirus carrying control and shRNAs targeting the indicated genes respectively. All data is normalized to Gapdh and shown relative to WT ESCs (set at 1.0). **b** The indicated cells (2000 cells per $cm^2$ in 6-well plates) were cultured for six days and cell numbers were counted. **c** The morphology of colonies formed by the indicated cell lines. Cells (1000 or 2000 cells in 48-well plates) were grown for 10 days. Scale bar, 50 μm. **d** Quantitative analysis of the colony formation assay in the indicated cell lines. Experiments were repeated three times and a representative result is shown. Data in (**a**–**c**) represent the mean ± SD; $n = 3$.

included in our re-defined modules. Fourth, the issues of target gene expression levels in ESCs were addressed in the present study.

The PRC module mainly includes development associated genes and is generally repressed in self-renewing ESCs[54]. However, increased PRC module activity was observed in E2.5-E4.5 pluripotent cells. Consistently, the same activity pattern was also shown for the PRC-Kim module. These results suggest an unexpected role of this functional module during early embryo development. We also noted that many target genes were shared by different modules, suggesting that these genes may be controlled by multiple mechanisms or that different modules may converge and co-operate to regulate specific biological processes in ESCs.

The activity of the newly defined PAF module is high in ESCs. Despite sharing some targets with the MYC and CORE modules, the PAF module was found to be specifically involved in nonsense mediated decay (NMD), regulation of translation and ribonucleoprotein complex assembly. Moreover, as previously mentioned, the Paf1 complex (the key factor of the PAF class) can promote the transcription of core factors (Oct4, Nanog and Sox2) by positively regulating enhancer activity[55]. Meanwhile, the Paf1 complex can also influence the transcription of Myc target genes

by interacting with Myc[56]. These results suggest that, importantly, PAF may also function in connecting multiple modules in the regulatory network.

ESCs and cancer cells have been reported to have similar gene expression signatures[57]. However, the precise nature of the gene expression regulation mode shared by ESCs and cancer cells still needs to be clarified. Kim et al[7] found that only the MYC module accounts for the similarity of transcriptional programs between ESCs and cancer cells. Given that cancer cells and ESCs share many cellular and molecular features, several core pluripotency genes, such as Oct4, Sox2, Klf4 and Nanog, are highly expressed in various types of human cancers, and some "NOS target gene sets" are enriched in high-grade tumours[47,58].It could be speculated that other gene modules may reflect stemness in cancer. In this study, using the newly defined modules as analytical tools, we observed that at least the CORE and PAF modules, as well as the MYC module were all shared by ESCs and cancer cells. Additionally, we observed different module activity patterns between different tumour sub-types. As it has become a consensus that cancer stem cells are responsible for the initiation and progression of tumours, it is urgent to discriminate and establish the stemness gene regulatory network of different types of tumours.

In conclusion, based on a functional genome-scale CRISPR screen and integrative analysis of multi-omics data, we re-defined the PGRN in ESCs. The newly constructed PGRN contains more comprehensive gene classifications and precise regulatory relationships. Whether and how these functional modules are coordinated to maintain the self-renewal of ESC needs to be further investigated. Nonetheless, these results will improve our understanding of the molecular mechanisms of pluripotency maintenance. Along with the advancement of cell tracing and bioinformatics technology, it is also anticipated that elaborate regulatory networks in other stem cells will be constructed.

## Materials and methods

**Plasmid construction**. All sgRNA and shRNA targeting sequences (Supplementary Tables 2, 3) and primers (Supplementary Tables 4, 5) were designed and BLASTed to ensure specificity. The sgRNAs were cloned into the pLentiCRISPR v2 vector (Addgene plasmid # 52961)[59]. The lentiCas9-Blast was a gift from Feng Zhang (Addgene plasmid # 52962)[59], and the Brie pooled library was a gift from David Root and John Doench (Addgene plasmid #73633)[20]. The shRNAs were cloned into the pLL3.7zeo vector[60].

The full-length ORFs of Tbx3, Ctr9, Pcgf6 and Dpf2 were PCR amplified from mouse ESC cDNA using KOD-Plus- (TOYOBO) and cloned into pGEM-T Easy (Promega). After DNA sequence verification, the ORFs were subcloned into pPyCAGIH[60]. The 3×FLAG fragments were obtained via PCR, and subcloned in-frame into the expression vectors[39].

**Lentiviral production**. Viral particles were produced in 293FT cells by co-transfection with the lentiviral vector, pSPAX2 and pMD2G (4:3:1) by calcium phosphate transfection and concentrated by ultracentrifugation at 70,000 g for 2 h[60].

**Cell lines and culture conditions**. Mouse ESC line R1 (American Type Culture Collection) was cultured on plastic dishes precoated with 0.1% (w/v) gelatine with Dulbecco's modified Eagle's medium (DMEM), supplemented with 5% ES cell-qualified foetal bovine serum (FBS), 15% KSR, 2 mM GlutaMAX, 1 mM sodium pyruvate, 0.1 mM non-essential amino acids, 0.1 mM β-mercaptoethanol (all from Invitrogen) and 10 ng ml$^{-1}$ LIF (Millipore)[60]. Cancer cell lines A549 and U87 (Cobioer) were cultured with Dulbecco's modified Eagle's medium (DMEM), supplemented with 15% FBS. The cells were routinely propagated by trypsinization and re-plated every two to three days.

**Generation of Cas9 expression cells**. Stable Cas9-expressing R1 cells (Cas9/R1) were generated by infecting R1 cells with lentiviral vectors containing lentiCas9-Blast and selected with blasticidin (5 µg ml$^{-1}$) for 7 days. The resistant colonies were pooled and expanded for further analysis.

**Functional screen using the Brie library**. As shown in Fig. 1a, Cas9/R1 cells were transfected with lentiviral vectors containing the Brie library at an MOI = 0.4. The cells were selected with puromycin (2 µg ml$^{-1}$) for 2 days and then propagated in L/S culture supplemented with puromycin for 14 days. The cells were passaged every 2 days. At every passage, approximately $4.0 \times 10^7$ cells were seeded in new tissue culture plates (~400 cells per sgRNA). A total of $4.0 \times 10^7$ cells were retained on day 0 and day14 post-screening for sequencing.

**DNA extraction, amplification, sequencing and statistical analysis**. Genomic DNA (gDNA) was isolated using Blood & Cell Culture Mini Kits according to the manufacturer's protocol (Qiagen). The extracted DNA and plasmid library were amplified as previously described[12]. For each DNA sample from cells, we performed 30 separate 100 µl reactions with 7 µg genomic DNA in each reaction using KOD FX DNA Polymerase (TOYOBO) and then combined the resulting amplicons. Primer sequences to amplify the sgRNAs were Crispr seqF: 5'- TTGTGGAA AGGACGAAACACCG-3' and Crispr seqR: 5'- TCTACTATTCTTTCCCCTG-CACTGT-3'. The PCR products were purified using a gel extraction kit (Omega). Samples were sequenced on a NextSeq machine (Illumina) at BIOZERON Co., Ltd. (Shanghai, China). Reads were counted by first locating the CACCG sequence that appears in the vector 5' in all gRNA inserts. The next 20 bases are the gRNA insert, which were then mapped to a reference file of all possible gRNAs present in the library using Bowtie 2.3.4.1[61]. Positive and negative selection genes were analyzed using MAGeCK software[62] with a threshold of $p$ value < 0.05.

**Chromosome localization**. Chromosome localizations of genes of interest were retrieved from the NCBI database (https://www.ncbi.nlm.nih.gov/), in which the number of genes at different chromosome regions is available. The R package (RIdeogram)[63] was used to display this information.

**Sub-cellular localization**. Localizations of proteins into cellular compartments were retrieved from the GO cellular component data source of the g:Profiler website (http://biit.cs.ut.ee/gprofiler/gost)[64]. The cellular components were divided into 9 compartments according to the sub-cellular localization: nucleus, mitochondrin, golgi, cytosol, ribosome, vesicle, peroxisome, endoplasmic reticulum. If one gene was enriched in multiple cellular components, we assigned it to all the associated compartments equally.

### Functional enrichment analysis

*Functional categorization of protein.* Functional categorization of the negative selection genes was performed using the PANTHER website (http://www.pantherdb.org/)[65].

*Gene Ontology.* Molecular function (MF), cellular component (CC) and biological process (BP) analyses were performed by using the g:Profiler website[64]. GO terms significant at an FDR-corrected $p$ value <0.05 were summarized (Supplementary Data 1), and the top 10 terms are shown. The detailed BP enrichment results are presented as a network diagram using the Cytoscape app 'Enrichment map'[66,67].

*Signalling pathways.* Curated pathways categories of the Gene Set Knowledgebase (GSKB, https://bioconductor.org/packages/gskb/) were identified by using GSEA software[68].

**Comparing the screening results with previously published datasets**. The raw data from four screening studies were obtained from the supplemental of the references (Supplementary Data 2). To exclude any bias caused by different analysis methodologies, we re-analysed the raw data from the shohat[26], Tzelepis[13] and Zhao[24] studies with MAGeCK[62]. For Li's research[25], we used the result (GFPplus-d15_vs_Plasmid. txt) calculated from MAGeCK provided by the author directly. The normalized results are summarized in Supplementary Data 3. Pearson correlation analysis was performed based on the normalized $\log_2$-fold change (LFC) of genes in each screening result.

**WGCNA**. WGCNA[69] was performed to construct a gene co-expression network across different samples, including ESCs, EB differentiation and three germ layer directional differentiations. First, to calculate the adjacency matrix, weighted co-expression relationships were evaluated by using paired Pearson correlations. Second, we converted the adjacency matrix into a topological overlap matrix (TOM). Then, genes with high expression correlations were clustered into modules based on TOM. Finally, a heatmap was drawn to show the correlation between different co-expression modules and samples. The data used for WGCNA were RPKM standardized format and all downloaded from public database. Among them, the data describing the gene expression of EB differentiation (GSE120224)[70] was downloaded from the GEO database. The RNA-seq data of three germ layer directional differentiation samples (E-MTAB-4904)[71] was downloaded from the BioStudies database (https://www.ebi.ac.uk/biostudies/). The RPKM processed expression data of mESCs (GSE53387) were from the supplemental data of reference[72,73].

**Cell proliferation and colony formation assay**. For the proliferation assay, the mESC cells were plated at a density of 1000 cells per cm$^2$ in gelatine-coated 12-well plates and cultured for 4 days, the cancer cells were plated at a density of 2000 cells per cm$^2$ in 6-well plates and cultured for 6 days. Viable cells were determined by Trypan blue exclusion and counted with an automated cell counter (Countstar BioTech).

For the colony formation assay, ESCs were plated at clonal density (50 cells per cm$^2$) and cultured for 6 days. Colonies were stained with a BCIP/NBT AP detection kit (Beyotime) according to the manufacturer's instructions and scored in three categories: undifferentiated, mixed (partially differentiated) and differentiated[60]. Cancer cells were plated at a density of 1000 or 2000 cells per well in 48-well plates and cultured for 10 days.

**Enrichment analysis of the iSRGS**. The enrichment analysis of the iSRGS was performed using GSEA[68]. GSEA has a robust ability to find more consistent results from independent datasets obtained with different platforms than from a single dataset[68]. We further performed GSEA on the five screen results individually. Enrichment pathways that met any of the requirements were selected: (1) the enrichment terms in the iSRGS with a $q$-value <0.05; (2) the enrichment terms in the iSRGS with a $q$-value ≥0.05 but were enriched in more than 2 out of the 5 screen results with $q$-value <0.2; and (3) the enrichment terms in the iSRGS with a $q$-value ≥0.05 but were enriched in more than 3 out of the 5 screen results with a $q$-value <0.5 (Supplementary Data 6). The detailed enrichment results are presented as a network diagram using the Cytoscape app 'Enrichment map' (Fig. 4A)[65,67].

### Classifications of the sub-classes in the PGRN based on co-occupancy

*ChIP-seq data download and processing.* A total of 374 ChIP-seq datasets (Supplementary Data 7) for 145 transcriptional regulators (59 genes in the known

network[31] and 86 newly identified genes in iSRGS) were downloaded from the public database. All the data were from mESCs cultured under L/S conditions. ChIP-seq data were downloaded from the Sequence Read Archive (SRA) in the National Center for Biotechnology Information (NCBI) database. FASTQ files were extracted from SRA files by SRA-Toolkit 2.9.2 (https://hpc.nih.gov/apps/sratoolkit.html). FASTQ quality was checked by FastQC (https://www.bioinformatics.babraham.ac.uk/projects/fastqc/), and low-quality bases of FASTQ were removed by Trim_Galore 1.18 (https://www.bioinformatics.babraham.ac.uk/projects/trim_galore/). FASTQ files were aligned onto the mouse genome (mm10) using Bowtie 2.3.4.1[61]. Binding sites of DNA-binding proteins and histone marks were both identified by model-based analysis of ChIP-seq peak caller (MACS) 2.1.4[74]. For each peak calling, if one experiment had control ChIP-seq data, the control data were used to remove the background noise; if the experiment was without control data, we chose GSM307154[75] as universal control data. For technical replicates in the same laboratory, only intersection regions of peaks from all replicates were used based on the intersect command in BEDTools software[76].

*Gene co-occupancy.* The degree of co-occupancy between two genes was calculated by the Z-score value as previously described[31]. The Z-score matrix is presented as a heatmap. A total of 126 genes (with 171 ChIP-seq datasets) were clustered into nine sub-classes (Fig. 5a), whereas the other 19 genes were excluded because of low co-occupancy degrees. Of note, the P53 and REST classes were included in the nine classes as previously reported[31] despite low co-occupancy degrees. The genomic annotation and distance distribution of each peak file were visualized by using the ChIPseeker R package[77] (Supplementary Fig. 3a).

*Acquisition of co-occupancy binding sites.* To acquire high-degree co-occupancy binding sites (HDBS), we first merged all the binding sites of the genes in one sub-class using the merge command in BEDTools software. Then the intersection command was used to calculate the number of genes occupied at each site. The sites with the top co-occupancy degree were selected[76]. In total, the CORE class targets were identified as sites co-occupied by 25 of 32 factors, the MYC class targets were identified as sites co-occupied by 36 of 43 factors, the PRC class targets were identified as sites co-occupied by 11 of 12 factors, the CTCF class targets were identified as sites co-occupied by 7 of 7 factors, the PCGF class targets were identified as sites co-occupied by 4 of 5 factors, the PAF class targets were identified as sites co-occupied by 5 of 6 factors, the TBX class targets were identified as sites co-occupied by 12 of 20 factors, the P53 class targets were identified as sites co-occupied by 1 of 1 factor, the REST class targets were identified as sites co-occupied by 1 of 1 factor(Supplementary Data 8).

**Identification of the histone modification status.** The RPGC of ten histone marks was normalized by using bamCoverage tools in deepTools software (3.1.3)[78]. The chromatin modification degrees on each co-occupancy binding site were calculated using computeMatrix tools in deepTools software[78]. Visualization was realized by using plotHeatmap tools in deepTools software[78].

**Defining the target genes of the sub-class**
*Predicting putative target genes.* The proximal target genes (TSS ± 5 kb) were obtained by using the ChIPseeker R package's seq2gene function[77]. The chromatin interactions between distal regulatory elements (REs) and promoters were determined based on a published Capture Hi-C (CHi-C) dataset in ESCs[79]. Genes with distal REs occupied by the sub-class regulators were considered putative distal target genes. The proximal and distal target genes were combined as putative target genes of the sub-class.

*Acquisition of the final target genes.* AdaSampling was used to optimize the putative target genes as previously reported[34,35]. To reduce the false-positive rate (FPR), we set the AdaSampling prediction threshold to 1. Histone modifications were quantified for each gene by calculating the RPKM in 5 kb region around the TSS. Finally, the sub-class target genes with similar transcriptional patterns and epigenomic features were discriminated based on the expression profiles (time-course RNA-seq data of EB formation) and epigenomic data (ChIP-seq data of 10 histone markers) (Supplementary Data 9).

**Chromatin immunoprecipitation (ChIP).** ChIP was performed using the ChIP Assay Kit (Millipore) according to the manufacturer's instructions. DNA fragments with an average size of 500 bp were immunoprecipitated using anti-Flag (Beyotime) monoclonal antibodies. Quantitative PCR analyses were performed using the Eco real-time PCR System (Roche) and SYBR green master mix (Roche). All the ChIP-qPCR primers are listed in Supplementary Supplementary Table 5.

**RNA-seq.** Total RNA was extracted using Trizol Reagent according to the manufacturer's manual. RNA samples were sent to Sangon Biotech Co., Ltd. (Shanghai, China) for mRNA sequencing on Illumina Hiseq platform (Illumina) with 6 Gbps.

**Module expression activity analysis.** RNA-seq datasets were collected and processed as follows. The RNA-seq data of hESCs (GSE143371)[80] were downloaded from GEO. The RNA-seq data of lung cancer and gliomas as well as the clinical survival data of gliomas were downloaded from the Xena dataset (https://xenabrowser.net/datapages/)[81]. SRA file download, FASTQ extraction and quality control processes of RNA-seq data were the same as those of ChIP-seq data, as described before. FASTQ files were aligned to the mouse genome mm10 using HISAT2 2.1.0[82]. We used featureCount software to calculate gene expression from aligned bam files[83]. First, gene expression levels were calculated as counts of the exon model per million mapped reads (CPM) to remove interference from sequence depth. Second, the mRNA expression of a gene *i* was standardized across all the samples:

$$S(i,j) = \log 2 \frac{G(i,j)}{\bar{G}(i)} \qquad (1)$$

where for each gene $G(i,j)$ is the expression level of mRNA at sample *j* and $\bar{G}(i)$ is the average expression of each gene *i* across all the samples. After standardization, the centralized expression of genes $C(i,j)$ at sample *j* was calculated as the centralized and standardized expression values at each sample. Positive values represent up-mean expression and vice versa.

$$C(i,j) = S(i,j) - \bar{S}(i) \qquad (2)$$

To quantify the average expression activity of the module, we first extracted the local data $C(i_x, j_y)$ from the total data $C(i,j)$, where the gene $i_x$ belongs to the specified module (x ∈ *module*), and sample $j_y$ belongs to the specified tumour pathological sub-type (y ∈ *subtype*). Next, we summed the local data $C(i_x, j_y)$ and divided this by summation of all the genes $g_x$ in all the samples $s_y$, where $g_x$ and $s_y$ are the total number of genes $i_x$ and total number of samples $j_y$, respectively. The data are represented by a bar graph to show the average expression activity of the module.

$$\begin{cases} i_x = i(x \in module) \\ j_y = j(y \in subtype) \\ E(x,y) = \sum_{1 \le i_x \le g_x} C(i_x, j_y) \cdot \frac{1}{g_x \cdot s_y} \\ 1 \le j_y \le s_y \end{cases} \qquad (3)$$

**Mapping the module activity patterns in early embryos.** To analyze the module activities in E2.5-E7.5 mouse embryos, we downloaded the single cell RNA-seq data from public database (E2.5-E4.5: E-MTAB-2958 and E-MTAB-2959[41]; E5.5-E7.5: GSE120963[40]), the activity score of each module in each sample was computed using AUCell (https://bioconductor.org/packages/release/bioc/vignettes/AUCell/inst/doc/.html) as previously described[40]. Finally, corn plots were generated based on the activity score of each module.

Spatial coordinates in the corn plot are as follows: anterior (A); posterior (P); left lateral (L); right lateral (R); anterior left lateral (L1); anterior right lateral (R1); posterior left lateral (L2); posterior right lateral (R2); divided epiblast (Epi1 and Epi2); whole mesoderm (M); anterior mesoderm (MA); posterior mesoderm (MP); divided endoderm (En1 and En2); anterior endoderm (EA) and posterior endoderm (EP).

**Survival analysis.** All the clinical survival data of cancer were downloaded from the public databases. The patient samples of each cancer were divided into the top 50% and bottom 50% groups according to the module activities. Then, the survival R package was used to do the Kaplan–Meier analyses of the progression-free interval in patients. *p* values are calculated using the log-rank test.

**Analysis of gene set enrichment pattern.** The gene set enrichment patterns were analysed as previously reported[46]. Briefly, for each sample, we first scored the genes whose expression was at least 2-fold over or under the average expression level and defined them as over- or under-expressed gene sets. Next, we assessed the fraction of over- and under-expressed genes in each sample belonging to a particular module and calculated the *p* value based on hypergeometric distribution. We used a threshold of $p < 0.05$ for significant enrichment. If both the over- and under-expressed genes were significantly enriched, we chose the gene set with the smaller *p* value and displayed it in the heatmap. Third, patient samples were clustered into different groups by clinical classification criteria. We assessed the fraction of samples enriched in the over- or under-expressed gene sets belonging to a particular group and calculated the *p* value based on hypergeometric distribution. The normalized expression data files and sample annotations were found at http://jura.wi.mit.edu/bioc/benporath/[46].

**Statistics and reproducibility.** The statistical analyses of the data are noted in the respective section describing the methods details. The figure legends give full information about the number of independent biological replicates (*n*) analyzed.

**Reporting summary**. Further information on research design is available in the Nature Portfolio Reporting Summary linked to this article.

## Data availability

The raw data of the four screening studies, Shohat (2019)[26], Tzelepis (2016)[13], Li (2018)[25], and Zhao (2017)[24], were downloaded from the supplementary material of these references. RNA-seq data were downloaded from GEO with accession number GSE120224 for EB differentiation of mESC (0d versus EB12d)[70] (Fig. 2e; Fig. 7a); GSE143371 for EB differentiation of hESC[80] (0d, 5d,10d) (Fig. 7d); and GSE44067[22] (Fig. 1i; Fig. 2c) and GSE53387[73] for mESCs cultured in LIF/serum. RNA-seq data were downloaded from EMBL-EBI with accession number E-MTAB-4904 for three germ layer directional differentiation of mESCs[71] (Fig. 2e; Fig. 7b) and E-MTAB-2958 and E-MTAB-2959 for E2.5-E4.5 mouse embryos[41] (Fig. 7c). scRNA-seq data were downloaded from GEO with accession number GSE65525 for mESCs cultured with LIF/serum[27] (Supplementary Figure 2d); GSE116165 for mouse E4.5 embryos[28] (Supplementary Figure 2c) and GSE120963 for mouse embryo spatiotemporal scRNA-seq (E5.5-E7.5)[40] (Fig. 7c). Microarray data were downloaded from GEO with accession number GSE3749[84] for mESC EB differentiation (Fig. 4b), GSE4189[85] for mNanog KD mESCs, and GSE27881[86] for mMax KD mESCs. Microarray data of mSuz12 KD mESCs was downloaded from the supplementary material of reference[87] (Supplementary Figure 7e). mESC histone ChIP-seq data were downloaded from GEO with accession number GSE11724 for H3K79me2[75]; GSE12241 for H4K20me3[88]; GSE24164 for H3K27ac[89]; GSE25532 for H3K27me3[90,91]; GSE27827 for H3K4me2[92]; GSE29218 for H3K4me3[93]; GSE29413 for H3K9me3[94]; GSE30203 for H3K4me1[95]; GSE31284 for H3K9ac[96]; and GSE41589 for H3K36me3[97]. RNA-seq data for lung cancer and gliomas as well as the clinical survival data of gliomas were downloaded from the Xena dataset (https://xenabrowser.net/datapages) (Fig. 8a–c; Supplementary Figure 5a-b). The breast and bladder normalized expression data files as well as sample annotations were found at http://jura.wi.mit.edu/bioc/benporath/ (Supplementary Figure 5c–f). The ChIP-seq data for transcriptional regulators were downloaded and are listed in Supplementary Data 7. RNA-seq data for Tbx3 KD, Ctr9 KD and Pcgf6 KD mESCs have been deposited in the GEO under the accession numbers GSE219206.

## Code availability

The scripts used for data analysis were uploaded to github (https://github.com/Jack123-Wang/A-multi-omicsRuan).

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

## Acknowledgements

We are grateful to Professor Yan Hu and Professor Hongli Li for their support with our research. This work was supported by grants from the National Natural Science Foundation of China (81702294), Joint Project of Chongqing Science and Technology Commission (2020MSXM054) and New Technology Cultivation Project of Military Medicine (CX2019JS202).

## Author contributions

R.J., Y.R., and J.Q.W. designed the project. J.Q.W., M.Y., F.S.W., and J.J.W. performed the bioinformatics analysis. Y.P.T., J.L.Z., R.Y., L.L.L., Y.D.C., Y.X.X., C.Z., Y.Y., and M.Y. performed the experiments. Y.R. and R.J. wrote the manuscript. R.J. conceived and supervised the study. G.X.C., J.L.W., W.W., and Y.H. contributed to the analysis and interpretation of data.

## Competing interests

The authors declare no competing interests.

## Additional information

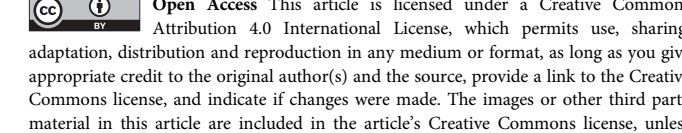

