## [Peer Review File · Communications Biology]

Reviewers' comments:

Reviewer #1 (Remarks to the Author):

The authors describe a CRISPR screen in mouse embryonic stem cells using a third-party library, which they followed up by integrating their results with data from previous works. This approach aimed at identifying gene regulatory clusters essential for stem cell function and maintenance. They then examined the resulting regulatory clusters across different publicly available datasets and describe their different properties towards stem cell function. The manuscript is well written and up to scientific standards. The data is reported transparently except some data not shown that should be added to the manuscript upon further processing. The figures are well designed and structured. The premise of the study is valid and the authors do efforts to integrate their data with previous screening results, yet reveal great variability between studies. This should be further discussed as replication studies are of great value to the scientific community. Yet some of the descriptive conclusions how the modules relate to different genomic features remain speculative. Functional validation of at least the newly identified clusters should be considered. Before publication, the newly defined PGRN sub-clusters need functional validation that goes beyond essentiality and stress measurements to allow conclusions about their impact on stem cell behavior.

Major comments

+ from the GSEA analysis it seems that the hit list is strongly enriched for core-essential genes (see Hart et al. 2015). These (e.g. Ribosome or OxPhos related genes) should be filtered out of the hitlist as they are presumably not related to stem cell maintenance.

+ thus, one could argue that some of the hits are rather needed for general cell fitness rather than "maintenance of self-renewal in mESCs" as the authors claim.

+ all data should be shown, e.g. the qPCR after sgRNA knockout of candidate genes

+ ESC fitness and identity seem to be used confusingly. Indeed, the CRISPR screen does not allow to infer about genes and their impact on cell identity but only about their essentiality for survival and proliferation.

+ it is at times very hard to distinguish if novel data produced in this study was used for analysis or if public data was used. Either explain the experimental rationale and method better in the result or introduce public datasets and their sources better.

Minor comments

+ results in hESC: what is meant with activity pattern? Did the authors mean expression dynamics?

+ regarding "context-specific" target genes, it remains elusive what the contexts could be and how they relate to the gene sets in question

+ scripts used for analysis or a link to a script collection on github, for example should be added

Reviewer #2 (Remarks to the Author):

The authors propose an extended list of gene regulatory network in mouse embryonic stem cells based on combined analysis of one new CRISPR screen and data from previously-published screens performed in comparable systems. The computational analyses are thorough and

comprehensive; however, statements would benefit from more in-depth experimental validation, i.e. generation and RNA-seq of KO or shRNA cell lines for selected components of the newly-identified modules, and ChIP-seq of selected factors of the newly defined PCGF, PAF, TBX modules. The authors finally extend their observation to cancer, correlating the expression of certain modules with prognosis/survival in specific type of gliomas and other cancers. It would be interesting/beneficial to corroborate those findings with functional validation in cancer cell lines.

MINOR POINTS.

1) Page 4: The abundances of the sgRNAs targeting non-/lowexpressed genes ($FPKM \leq 0.5$) remained the same as the initial pool (P.Sc_0d) (Fig. 1e), indicating that the off-target effects are negligible.

This statement only applies to sgRNA targeting non-low expressed genes, it is not possible to generalize this the full sgRNA library, please rephrase or remove.

2) Page 4/5: Using MAGeCK [21], we detected 2930 negative selection genes and 1384 positive selection genes 5 with p value < 0.05 (Supplementary Data 1).

This sentence is not clear. What do the author mean with 'negative selection genes'? Please rephrase using a different terminology, i.e. using MAGeCK we detected 2930 genes whose sgRNAs were depleted, suggesting those as genes required for self-renewal, as well as 1384 genes whose sgRNAs were enriched, indicating genes...

3) Page 5: A total of 44.3% of the genes were located in the nucleus, 35.2% were located in mitochondria, and the rest were distributed in the cytosol, and among ribosomes and the cytoskeleton (Fig. S1b).

Please rephrase: 44.3% of the identified genes encoding proteins localized in the nucleus, 35.2%...

MAJORPOINTS

1) Page 6: The silencing level of each sgRNA was measured and confirmed by qRT-PCR (data not 7 shown).

Insertion/deletion introduced by CRISPR might not affect at all the RNA abundance of the target gene. Yet, indels introduce premature stop codons that lead to no/truncated protein. The author should provide a thorough analysis of their KO cells. i.e. by western-blot. Moreover, the author should describe if KO cells are a bulk population or if those are clonally-selected homozygous KO cells.

2) The authors should validate experimentally the new proposed modules by profiling their chromatin binding o chip-sequencing or similar methodology in mESC cells and differentiation.

2) Page 11: the authors report a correlation with gene expression of CORE/MYC/PAF modules in gliomas vs normal brain tissue and also a correlation with tumour grade. The author should validate those finding experimentally and go further by testing human cancer cell lines, either consulting available database (i.e. Achilles/DepMap) or by generating new data (i.e. KO components of CORE/MYC/PAF and assessing viability/cell cycle).

RESPONSE TO REVIEWERS –COMMSBIO-22-3745A

We thank the reviewers for their thoughtful and constructive review of our manuscript. We have endeavored to revise our manuscript according to the comments we have received. A list of the new added experimental data and a detailed point-by-point response are included below, with the reviewers' comments indicated in italics. All changes made in the text are marked in red so that they may be easily identified.

List of the added experimental data:

1. Cell lines:

1) 3Flag-mCtr9 OE_R1, 3Flag-mTbx3 OE_R1, 3Flag-mDpf2 OE_R1 and 3Flag-mPcgf6 OE_R1.

2) mCtr9 KD_NG5 (Nanog-GFP reporter mESCs), mPcgf6 KD_NG5, mTbx3 KD_NG5 and control_NG5.

3) hNANOG KD_A549, hMYC KD_A549, hCTR9 KD_A549, and control_A549; hNANOG KD_U87, hMYC KD_U87, hCTR9 KD_U87, and control_U87.

2. CHIP assay for Ctr9, Tbx3, Dpf2 and Pcgf6, and qPCR analysis of their targets.

3. Functional analysis in mESCs: cell counting, colony formation assay and FASC in mCtr9 KD_NG5, mPcgf6 KD_NG5, mTbx3 KD_NG5 and control/NG5.

4. RNA-seq analysis: mCtr9 KD_NG5 vs. control_NG5; mPcgf6 KD_NG5 vs. control_NG5; mTbx3 KD_NG5 vs. control_NG5.

5. GO analysis for DEGs in mNanog, mMax, mSuz12, mCtr9, mPcgf6 and mTbx3 KD

mESCs.

6. Functional analysis in cancer cells: cell counting and soft agar colony formation assay in control_A549, hNANOG KD_A549, hMYC KD_A549, hCTR9 KD_A549, control_U87, hNANOG KD_U87, hMYC KD_U87 and hCTR9 KD_U87.

Point-by-point responses to the reviewers' comments:

Reviewer #1

- Major comments

1. *From the GSEA analysis it seems that the hit list is strongly enriched for core-essential genes (see Hart et al. 2015). These (e.g. Ribosome or OxPhos related genes) should be filtered out of the hitlist as they are presumably not related to stem cell maintenance. Thus, one could argue that some of the hits are rather needed for general cell fitness rather than "maintenance of self-renewal in mESCs" as the authors claim.*

- As suggested by the reviewer, we have tried to remove core-essential genes (defined by Hart et al.)¹ from the self-renewal related gene set (iSRGS). After doing so, a total 39 genes would be excluded from the new constructed PGRN (shown as below). However, we found almost all the filtered genes in PGRN were those that have been validated to play key roles in mESC pluripotency maintenance (related references were listed in Supplementary Data7). Moreover, these genes constituted the important integrants of the transcriptional regulation sub-class (Fig 5).

Additionally, the GSEA in the present study revealed that those genes regulating basic cellular functions, such as “oxidative phosphorylation”, “ubiquitin proteasome”, “mRNA processing” and “translations” pathways, were highly expressed in mESCs and downregulated after differentiation (Fig 4). These results were in accordance with previous studies²⁻⁴, which showed that specific metabolism processes were intimately linked to the pluripotent state of ESCs, indicating that some fundamental cellular pathways may also be necessary and specific for stem cell maintenance. Therefore, we finally retained these essential genes to make the iSRGS and PGRN more comprehensive.

References

- 1) Hart, T. et al. High-Resolution CRISPR Screens Reveal Fitness Genes and Genotype-Specific Cancer Liabilities. *Cell* 163, 1515-1526 (2015).
- 2) Naik, P.P. et al. Mitochondrial Heterogeneity in Stem Cells. *Adv Exp Med Biol* 1123, 179-194 (2019).
- 3) Gabut, M., Bourdelais, F. & Durand, S. Ribosome and Translational Control in Stem Cells. *Cells* 9 (2020).
- 4) Li, D. & Wang, J. Ribosome heterogeneity in stem cells and development. *J Cell Biol* 219 (2020).

2. All data should be shown, e.g. the qPCR after sgRNA knockout of candidate genes.

- We have added these data to a revised version of Fig. S3 as suggested by the reviewer.

3. ESC fitness and identity seem to be used confusingly. Indeed, the CRISPR screen does not allow to infer about genes and their impact on cell identity but only about their essentiality for survival and proliferation.

- We appreciated for your reminding on this important point. In the revised version of this manuscript, we have modified these sentences (line 358, 373 and 432) in discussion to avoid ambiguity.

4. It is at times very hard to distinguish if novel data produced in this study was used for analysis or if public data was used. Either explain the experimental rational and method better in the result or introduce public datasets and they sources better.

- We have modified the descriptions of methods and indicated the public dataset sources accordingly to make the point clearer (Page 18-23).

5. Before publication, the newly defined PGRN sub-clusters need functional validation that

goes beyond essentiality and stress measurements to allow conclusions about their impact on stem cell behavior.

- In the revision version, we performed further functional analysis to validate the impact of newly defined PAF, PCGF and TBX3 modules on mESCs self-renewal. The major transcriptional factors of each module (Ctr9, Pcgf6 and Tbx3) were knocked down individually by RNAi in a Nanog-GFP reporter ESC line (Fig. S7a). Compared to the control ESCs, each of the KD cells displayed a limited proliferation capacity (Fig. S7b). Colony forming assays revealed that Ctr9, Pcgf6 and Tbx3 KD cells formed fewer undifferentiated alkaline phosphatase (AP)-positive colonies (Fig. S7c), while fluorescence-activated cell sorting (FACS) analysis showed a significant decreased Nanog-GFP positive rate as well (Fig. S7d). RNA-seq was performed and GO analysis for DEGs in Ctr9, Pcgf6 and Tbx3 KD cells showed that the three modules were primarily involved in development regulation. Besides, the PAF and PCGF modules were shown to influence signaling transduction, while TBX and PCGF modules were involved in cell migration regulation (Fig. S7e). These results indicated that PAF, PCGF and TBX3 modules were all essential for mESC self-renewal but with distinct regulation mechanisms. This data has been added to a revised version of Fig. S7, and was described in Page 10.

- Minor comments

1. Results in hESC: what is meant with activity pattern? Did the authors mean expression dynamics?

- As the module activity was calculated by the average expression of all the genes in each module (described in methods, page 21), the activity pattern refers to the expression dynamics of the module genes during a time course of EB differentiation of hESCs.

2. Regarding “context-specific” target genes, it remains elusive what the contexts could be and how they relate to the gene sets in question.

- In this paper, the “context-specific” genes refer to genes that were identified by at least one but not present in all five analyzed screens. The discrepancy between these screenings may be due to the different contexts, such as sgRNA library, cell model, culture condition and analysis methods used by different studies (as we summarized in Table S1).

3. *Scripts used for analysis or a link to a script collection on github, for example should be added.*

- The scripts used for analysis were uploaded to github, and the link (<https://github.com/Jack123-Wang/A-multi-omicsRuan>) was added to the materials and methods section (page 24, line 683).

Reviewer #2

- Major comments

1. *Page 6: The silencing level of each sgRNA was measured and confirmed by qRT-PCR (data not shown). Insertion/deletion introduced by CRISPR might not affect at all the RNA abundance of the target gene. Yet, indels introduce premature stop codons that lead to no/truncated protein. The author should provide a thorough analysis of their KO cells. i.e. by western-blot. Moreover, the author should describe if KO cells are a bulk population or if those are clonally-selected homozygous KO cells.*

- We have added these data to a revised version of Fig. S3 as suggested by the reviewer.

2. *The authors should validate experimentally the new proposed modules by profiling their chromatin binding using chip-sequencing or similar methodology in mESC cells and differentiation.*

- We appreciated for your reminding on this important point. To validate the new proposed PAF, TBX3 and PCGF modules, we performed ChIP assay using anti-Flag antibody in

flag-tagged Ctr9 (for PAF module), Tbx3 (for TBX module), Dpf2 (for TBX module) and Pcgf6 (for PCGF module) transfected mESCs. Then 11 target genes in different modules were chosen for qPCR analysis. The results were consistent with previous reports¹⁻⁴ and showed that all the four transcriptional factors preferentially occupied the targets of their own modules, indicating the specific and reliable assignment of target genes for each module.

This data has been added to a revised version of Fig. S6, and was described in Page 10.

References

- 1) Han, J.Y. et al. Tbx3 improves the germ-line competency of induced pluripotent stem cells. *Nature* 463, 1096-+ (2010).
- 2) Rahl, P.B. et al. c-Myc Regulates Transcriptional Pause Release. *Cell* 141, 432-445 (2010).
- 3) Yang, C.S., Chang, K.Y., Dang, J. & Rana, T.M. Polycomb Group Protein Pcgf6 Acts as a Master Regulator to Maintain Embryonic Stem Cell Identity. *Scientific reports* 6 (2016).
- 4) Local, A. et al. Identification of H3K4me1-associated proteins at mammalian enhancers. *Nat Genet* 50, 73-82 (2018).

3. Page 11: the authors report a correlation with gene expression of CORE/MYC/PAF modules in gliomas vs normal brain tissue and also a correlation with tumour grade. The

author should validate those finding experimentally and go further by testing human cancer cell lines, either consulting available database (i.e. Achilles/DepMap) or by generating new data (i.e. KO components of CORE/MYC/PAF and assessing viability/cell cycle).

- In light of this comment, we have performed the functional validation of CORE, MYC and PAF modules in human cancer lines A549 and U87 by knocking down the main component of each module. The results are shown in Fig. 9 in the revised version of the manuscript and are described in the text (page 12) as follows: Furthermore, for assessing the function of CORE, MYC and PAF modules in cancer cells, NANOG (for CORE module), MYC (for MYC module) and CTR9 (for PAF module) were knocked down in glioma cell line U87 respectively (Fig. 9a). As expected, all of the three gene KD cells displayed a decreased proliferation rate alongside limited colony formation capacity when compared to the control cells (Fig. 9b-d). Consistently, the same phenotypes were observed in the respective gene KD lung adenocarcinoma A549 cells (Fig. 9). These findings supported previous analysis results on clinical tumor samples (Fig. 8), demonstrating the importance of the CORE, MYC, and PAF modules in cancer progression.

- Minor comments

1. Page 4: The abundances of the sgRNAs targeting non-/low expressed genes (FPKM \leq 0.5) remained the same as the initial pool (P.Sc_0d) (Fig. 1e), indicating that the off-target effects are negligible. This statement only applies to sgRNA targeting non-low expressed genes, it is not possible to generalize the full sgRNA library, please rephrase or remove.

- We have removed this statement in the revised version as suggested by the reviewer (Page 4, line 92).

2. Page 4/5: Using MAGeCK [21], we detected 2930 negative selection genes and 1384 positive selection genes with p value < 0.05 (Supplementary Data 1). This sentence is not

clear. What do the author mean with “negative selection genes”? Please rephrase using a different terminology, i.e. using MAGeCK we detected 2930 genes whose sgRNAs were depleted, suggesting those as genes required for self-renewal, as well as 1384 genes whose sgRNAs were enriched, indicating genes...

- We have rephrased this sentence as follows: “using MAGeCK, we detected 2930 genes whose sgRNAs were depleted, suggesting those as genes required for mESC fitness, as well as 1384 genes whose sgRNAs were enriched, indicating genes harmful to the self-renewal of mESCs” (Page 4, line 93-95).

3. Page 5: A total of 44.3% of the genes were located in the nucleus, 35.2% were located in mitochondria, and the rest were distributed in the cytosol, and among ribosomes and the cytoskeleton (Fig. S1b). Please rephrase: 44.3% of the identified genes encoding proteins localized in the nucleus, 35.2%...

- We have rephrased this sentence in the revised version as suggested by the reviewer (Page 5, line 100-102).

REVIEWERS' COMMENTS:

Reviewer #1 (Remarks to the Author):

Dear Authors,

After careful assessment of your revised manuscript I am delighted to state that my concerns have been addressed/discussed appropriately,

best

Reviewer #2 (Remarks to the Author):

In their revised manuscript and rebuttal, the authors satisfied the majority of my concern.